# The oncolytic adenovirus TILT-123 with pembrolizumab in platinum resistant or refractory ovarian cancer: the phase 1a PROTA trial

Immune checkpoint inhibitors have demonstrated modest efficacy as a monotherapy in ovarian cancer. Originally developed to improve efficacy of T-cell therapies such as immune checkpoint inhibitors and adoptive cell transfer, TILT-123 (Ad5/3-E2F-D24-hTNFα-IRES-hIL-2) is a serotype chimeric oncolytic adenovirus encoding tumor necrosis factor alpha and interleukin-2. Here we report results from phase 1a of PROTA, a single-arm, multicentre dose escalation trial with TILT-123 and pembrolizumab in female patients with platinum resistant or refractory ovarian cancer (NCT05271318). The primary endpoint was safety. Secondary endpoints included efficacy, tolerability, virus persistence and anti-viral immunity. Patients ($n = 15$) received intravenous and intraperitoneal and/or intratumoral injections of TILT-123 as well as intravenous pembrolizumab. Treatment was well tolerated, and no dose-limiting toxicities were observed. The most frequent adverse events were fever (40%), fatigue (40%) and nausea (40%). Disease control was achieved in 64% of evaluable patients (9/14). Median progression-free survival and overall survival were 98 and 190 days respectively. Clinical responses were associated with higher serum anti-adenovirus neutralizing antibody titer at baseline and post-treatment. The phase 1b investigating TILT-123, pembrolizumab and PEGylated liposomal doxorubicin in a similar patient population is underway.

Ovarian cancer (OC) is a leading cause of gynaecologic cancer deaths worldwide, and despite advances in surgery and chemotherapy, a high incidence of recurrence and treatment resistance occurs[1]. Accumulating evidence demonstrates that a proportion of high-grade serous ovarian cancer (HGSOC) are T cell inflamed, suggesting their suitability for immunotherapy[2]. However, single-agent immune checkpoint inhibitors (ICIs) have provided disappointing results in the clinic, with an objective response rate of only 9% with pembrolizumab monotherapy (KEYNOTE-100)[3]. Remarkably, patients benefiting from treatment have been observed, mainly in patients with high PD-L1 expression, thus providing an opportunity to optimize the therapy in a way that will increase the fraction of patients benefiting.

Oncolytic viruses (OVs) are distinguished by their properties to replicate in cancer cells, cause direct cell lysis, induce immunogenic cell death, and through genetic modification, express immune stimulating proteins, to further amplify an immune response[4,5]. In addition to local amplification of the antitumor effect, some types of OV are able to enter systemic circulation and reach distant metastases[6]. This includes chimeric serotype 5/3 oncolytic adenoviruses such as TILT-123 (Ad5/3-E2F-D24-hTNFα-IRES-hIL-2). Following encouraging clinical observations with older generation oncolytic adenoviruses in the

✉e-mail: akseli.hemminki@helsinki.fi

context of ovarian cancer[7], TILT-123 was used in this present study in combination with pembrolizumab.

TILT-123 is an oncolytic adenovirus encoding tumor necrosis factor alpha (TNFα) and interleukin-2 (IL-2), designed to complement T-cell therapies including immune checkpoint inhibition. The two cytokines were selected based on their ability to improve T-cell recruitment and activation in preclinical models of T-cell therapy[8–10]. Later studies demonstrated outstanding effects—even 100% cure rates —in vivo, when combined with anti-PD-1 or anti-PD-L1, in mouse and hamster models of ICI naïve or refractory melanoma, renal cell carcinoma, head and neck, ovarian, pancreatic and lung cancer[11–16]. Evaluation of mechanism of action has provided numerous insights including the critical role of NKs, CD4, and CD8 T cells, and the capability of TILT-123 to induce tertiary lymphoid structure formation in PD-L1 refractory tumors[13,16].

TILT-123 has also been shown to cause the release of more immunogenic danger associated molecular patterns and pathogen associated molecular patterns than single-armed or unarmed Ad5/3-E2F-D24 following infection[17]. The shift in antigenicity and adjuvanticity in the tumor microenvironment by TILT-123 is likely to induce local immunosuppressive feedback mechanisms, such as increased PD-1 and PD-L1 expression. Such mechanisms rationalise the use of TILT-123 with anti-PD-1 therapy, given the trend for association of PD-L1 expression with treatment benefits in KEYNOTE-100[3]. We hypothesize that TILT-123 can improve anti-PD-1 efficacy by immunologically inducing immunosuppressive and / or innate tumors while anti-PD-1 can improve TILT-123 efficacy by delaying T-cell exhaustion yielding an additive effect.

So far, there are no approved treatments using both OV and ICI, with notably disappointing results in a phase III trial with the oncolytic herpes virus encoding GM-CSF (T-VEC) and pembrolizumab in advanced melanoma[18,19]. In contrast to approaches utilizing GM-CSF, TILT-123 has been optimised by design to improve the efficacy of T cell therapies including ICIs[20,21]. Likewise, adenovirus may be advantageous over herpes regarding adaptive immunity[22].

In this article we report the results of phase 1a of PROTA, a single-arm, multicenter phase I dose escalation clinical trial of combined i.v. and i.t./i.p. injection of TILT-123, combined with systemic pembrolizumab for patients with platinum resistant or refractory ovarian cancer. Here we show the treatment is safe, well tolerated and can elicit disease control in a subset of patients despite extensive prior treatments. Preliminary biomarker analysis suggests the treatment induced antibody response is relevant for efficacy.

## Results

### Patient demographics and characteristics

A total of 15 patients were enrolled (between 18th May 2022 and 7th November 2023) with a median age of 66 (36–78) years, of which 14 were evaluable for treatment response under RECIST 1.1/iRECIST in at least one time point. The demographic and baseline clinical characteristics of all patients enrolled are reported in Table 1. Fifty-seven and forty-three percent of patients had a WHO/ECOG performance status 1 and 0 respectively. The prevalence of primary cancer type was epithelial ovarian cancer (60%), fallopian tube cancer (20%) and primary peritoneal cancer (20%) with the most prevalent histological subtype being HGSOC (73%). The percentage of platinum-refractory and -resistant cancers were 27% and 73% respectively. Patients received a median of 7 lines of systemic treatments prior to trial entry. A detailed description of patient diagnosis and previous lines of cancer treatments are presented in Supplementary Tables 1 and 2.

### Safety

Patients received an i.v. dose of TILT-123 on day 1, i.t. doses of TILT-123 on days 8, 22, 36, 57, and 78 and pembrolizumab (200 mg) infusion on days 36, 57 and 78. Doses for cohort 1 ($n = 3$), 2 ($n = 3$), 3 ($n = 3$) and 4

($n = 6$) were $3 \times 10^{11}$, $1 \times 10^{12}$, $2 \times 10^{12}$, $4 \times 10^{12}$ for i.v. injection and $1 \times 10^{11}$, $3 \times 10^{11}$, $3 \times 10^{11}$, $5 \times 10^{11}$ for both i.t. and i.p. injections. 14 out of 15 patients met first imaging time point at day 36 whilst 7 out of 14 patients met the primary endpoint at day 92 (Fig. 1a).

The majority of AEs related to the treatment were either grade 1 or 2 and the most common were fever (40%), fatigue (40%) and nausea (40%).

Treatment with TILT-123 and pembrolizumab was well tolerated and no dose-limiting toxicities were observed. The most frequent adverse events related to treatment were fever (40%), fatigue (40%) and nausea (40%). Two grade ≥ 3 adverse events related to treatment were reported and both were in cohort 4 (delirium and hemoperitoneum). There were no deaths that were related to treatment. An overview of adverse events related to treatment that occurred during the study are summarised according to grade and cohort in Table 2 and Supplementary Table 3 respectively.

No treatment related adverse events associated with complete blood count (with differential) were reported. An overview of dynamic changes to the blood compartment can be found in Supplementary Figs. 2 and 3. No signs of liver toxicity were noted across the trial, including following i.v. injection at the maximum dose, as measured by alanine aminotransferase (ALT), aspartate aminotransferase (AST), alkaline phosphatase (ALP) and lactate dehydrogenase (LDH) (Supplementary Fig. 3a–d). A non-significant increase in bilirubin was observed following infusion with pembrolizumab, however levels rapidly stabilised (Supplementary Fig. 3e). No significant changes in creatinine, potassium, sodium or thyroid stimulating hormone were observed (Supplementary Fig. 3f–i).

Three patients, two in cohort 1 and one in cohort 4 received i.p. doses of TILT-123 in the trial. Patients 302-05, 301-01, and 302-12 received three, four and two i.p injections (50 ml total volume per dose) respectively. 301-01 who presented with primary peritoneal cancer was the only patient to receive just i.p. injections during the trial, as the patient had no other injectable lesions. 302-05 and 302-12 received both i.p. and i.t. injections for treating ascites and other metastatic lesions respectively. No serious adverse events related to treatment were reported amongst these patients.

Analysis of adverse events related to i.t. injection included three grade 1 infusion site reactions related to TILT-123 in patients 301-02, 301-03, and 301-04 (Table 2). In addition, patients 301-03 and 302-07 reported two incidences of grade 2 abdominal pain on day 8 caused by activity of TILT-123. Three incidences of pain following injection site/ biopsy are reported in Supplementary Table 4 and were not related to treatment. Grade 3 hemoperitoneum was reported in patient 301-12, as related to injection of TILT-123, and not the activity of TILT-123. The patient required prolonged hospitalization but was eventually discharged in stable condition.

All adverse events (AEs) that occurred during the study are reported in Supplementary Table 4.

### Efficacy

The efficacy and survival endpoints are reported in Table 1. Median overall survival (OS) was 190 days (all patients) and median progression-free survival (PFS) was 98 days (Fig. 1b, c). Median time to progression (TTP) was 98 days (Supplementary Fig. 1a). The overall disease control rate by RECIST1.1 was 64% (9 out of 14 evaluable patients) with the best response being partial response (Fig. 1e, f). The overall response rate (ORR) was 7.1% (1/14) and 20% (1/5) at the highest dose level. 4/15 patients (302-06, 301-05, 301-11, 302-10) entered the extension treatment period and received a total of 22 additional treatments between them.

Treatment related tumor reductions were observed in both injected and non-injected tumors at both CT imaging time points (36 and 92) (Supplementary Fig. 1b). A reduction or no change in tumor diameter was observed in 6/15 (40%) and 9/16 (56%) injected and non-injected

**Table 1 | Patients' description and clinical characteristics**

| Patient ID | Cohort | ECOG status | Time since diagnosis (months) | Cancer type | Histological Subtype | Number of previous systemic treatments | Platinum status | ICI status | BRCA/HRD/MSI status | Best Overall Response RECIST 1.1 | Best Overall Response iRECIST | OS (days) | Serum anti-adenovirus antibodies at baseline |
|---|---|---|---|---|---|---|---|---|---|---|---|---|---|
| 302-05 | 1 | 1 | 63 | Fallopian tube cancer | HGSOC | 7 | Resistant | Naive | BRCA2mut/NA/NA | PD | iUPD | 85 | Negative |
| 302-06 | 1 | 1 | 282 | Epithelial ovarian cancer | LGSOC | 12 | Resistant | Naive | NA/NA/NA | SD | iSD | 306 | Positive |
| 301-01 | 1 | 1 | 30 | Primary peritoneal cancer | HGSOC | 4 | Resistant | Naive | Wt/Pos/Stable | SD | iSD | 77 | Negative |
| 301-02 | 2 | 1 | 13 | Epithelial ovarian cancer | HGSOC | 3 | Resistant | Naive | Wt/Neg/Stable | SD | iSD | 434 | Positive |
| 301-03 | 2 | 1 | 148 | Epithelial ovarian cancer | HGSOC | 12 | Resistant | Naive | Wt/NA/NA | SD | iSD | 122 | Negative |
| 301-04 | 2 | 1 | 62 | Epithelial ovarian cancer | HGSOC | 7 | Resistant | Resistant | Wt/NA/Stable | PD | iUPD | 93 | Positive |
| 302-07 | 3 | 1 | 26 | Fallopian tube cancer | HGSOC | 5 | Refractory | Naive | Wt/NA/NA | SD | iSD | 190 | Negative |
| 301-05 | 3 | 0 | 15 | Epithelial ovarian cancer | HGSOC | 1 | Resistant | Naive | Wt/Neg/NA | SD | iSD | 463* | Positive |
| 302-09 | 3 | 1 | 80 | Epithelial ovarian cancer | HGSOC | 8 | Refractory | Naive | Wt/NA/NA | PD | iUPD | 280 | Positive |
| 302-10 | 4 | 0 | 59 | Epithelial ovarian cancer | HGSOC | 11 | Resistant | Naive | Wt/Pos/NA | SD | iSD | 254* | Positive |
| 301-10 | 4 | 0 | 48 | Fallopian tube cancer | HGSOC | 5 | Resistant | Naive | Wt/Neg/Stable | PD | iUPD | 245* | Negative |
| 302-11 | 4 | 0 | 45 | Primary peritoneal cancer | LGSOC | 9 | Resistant | Naive | Wt/Pos/NA | NA | NA | 138 | Negative |
| 301-11 | 4 | 0 | 36 | Epithelial ovarian cancer | Mucinous carcinoma | 2 | Refractory | Naive | N/A/Neg/NA | PR | iPR | 224* | Negative |
| 301-12 | 4 | 0 | 20 | Epithelial ovarian cancer | Carcinosarcoma | 3 | Refractory | Naive | Wt/Neg/Stable | SD | iSD | 151 | Positive |
| 302-12 | 4 | 0 | 76 | Primary peritoneal cancer | HGSOC | 12 | Resistant | Naive | Wt/NA/N/A | PD | iUPD | 105 | Negative |

Description of 15 patients enrolled in PROTA who received treatment with TILT-123 and pembrolizumab. Patient identification number (ID), cohort, WHO/ECOG performance status, time since diagnosis (months), cancer type, histological subtype, number of previous systemic treatments, platinum status, ICI status, best CT response by RECIST 1.1 and iRECIST (either day 36 or 92) and adenovirus serostatus at baseline (negative and positive indicate lower and higher than 1:64 dilution respectively). HGSOC High-Grade Serous Ovarian Carcinoma, LGSOC Low-Grade Serous Ovarian Carcinoma, PD Progressive Disease, SD Stable Disease, iUPD immune Unconfirmed Progressive Disease, iSD immune Stable Disease. OS indicates overall survival with * still alive since enrolment. HRD Homologous Recombination Deficiency, Wt wild type (when both BRCA1 and BRCA2 were assessed and identified as wild type), Pos Positive, Neg Negative, NA not available. Data cut-off 16/05/24.

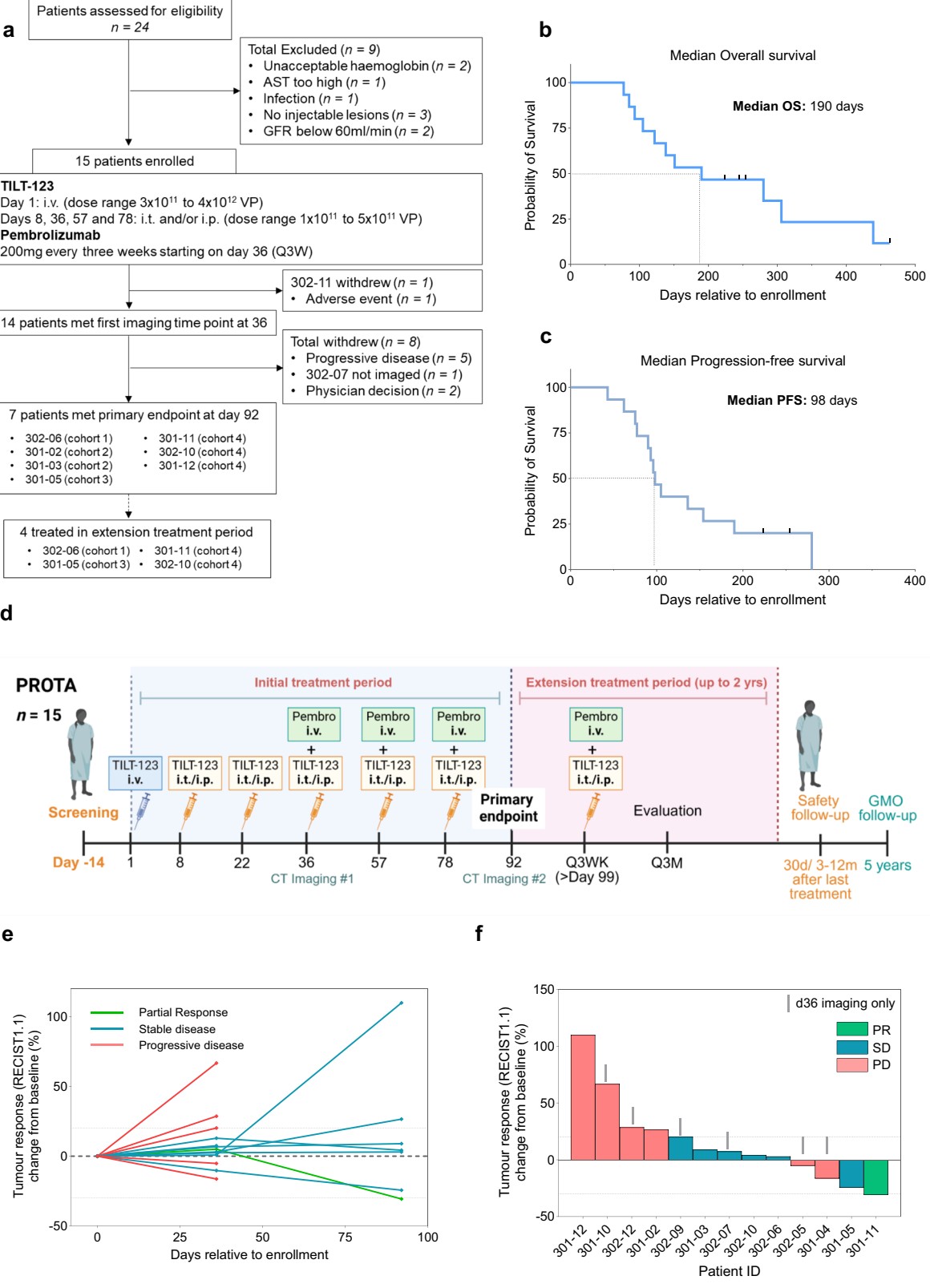

**Fig. 1 | Summary of treatment responses. a** Trial outline. **b** Median Overall survival. **c** Median Progression-free survival. **d** Treatment schedule (Created in BioRender. Clubb, J. [2024] https://BioRender.com/x81d347). **e** Spider plot showing sum of target-lesion response, displayed by best response (RECIST1.1) and evaluated by CT on day 36 and 92. **f** Waterfall plot showing sum of target-lesion response, displayed by response (RECIST1.1) at end of trial. Data cut-off 16/05/2024. Source data are provided as a Source Data file.

**Table 2 | Adverse events related to treatment as judged and reported by the investigator, stratified by grade and unique patients with reported adverse events**

| Event type | Grade 1 | Grade 2 | Grade 3 | Grade 4 | Grade 5 | Total | N unique patients experiencing AE (%) |
|---|---|---|---|---|---|---|---|
| Infection like symptoms | | | | | | | |
| Fever | 7 (47%) | 3 (20%) | 0 | 0 | 0 | 10 | 6/15 (40%) |
| Chills | 11 (73%) | 1 (7%) | 0 | 0 | 0 | 12 | 5/15 (33%) |
| Cough | 1 (7%) | 0 | 0 | 0 | 0 | 1 | 1/15 (7%) |
| Flu like symptoms | 2 (14%) | 0 | 0 | 0 | 0 | 2 | 1/15 (7%) |
| General | | | | | | | |
| Fatigue | 7 (47%) | 4 (27%) | 0 | 0 | 0 | 11 | 6/15 (40%) |
| Nausea | 10 (67%) | 0 | 0 | 0 | 0 | 10 | 6/15 (40%) |
| Headache | 3 (20%) | 0 | 0 | 0 | 0 | 3 | 3/15 (20%) |
| Delirium | 0 | 0 | 1 (7%) | 0 | 0 | 1 | 1/15 (7%) |
| Haematological | | | | | | | |
| Anaemia | 0 | 3 (20%) | 0 | 0 | 0 | 3 | 2/15 (14%) |
| Gastrointestinal | | | | | | | |
| Diarrhoea | 3 (20%) | 2 (14%) | 0 | 0 | 0 | 5 | 5/15 (33%) |
| Pyrosis | 1 (7%) | 0 | 0 | 0 | 0 | 1 | 1/15 (7%) |
| Vomiting | 4 (27%) | 0 | 0 | 0 | 0 | 4 | 3/15 (20%) |
| Anorexia | 6 (40%) | 0 | 0 | 0 | 0 | 6 | 5/15 (33%) |
| Abdominal discomfort | 2 (14%) | 2 (14%) | 0 | 0 | 0 | 4 | 2/15 (14%) |
| Abdominal pain | 2 (14%) | 1 (7%) | 0 | 0 | 0 | 3 | 3/15 (20%) |
| Hemoperitoneum | 0 | 0 | 1 (7%) | 0 | 0 | 1 | 1/15 (7%) |
| Respiratory | | | | | | | |
| Dyspnoea | 2 (14%) | 0 | 0 | 0 | 0 | 2 | 2/15 (14%) |
| Local symptoms | | | | | | | |
| Infusion site reaction | 3 (20%) | 0 | 0 | 0 | 0 | 3 | 3/15 (20%) |
| Alopecia | 2 (14%) | 0 | 0 | 0 | 0 | 2 | 2/15 (14%) |
| Skin rash | 1 (7%) | 1 (7%) | 0 | 0 | 0 | 2 | 1/15 (7%) |
| Total | 67 | 17 | 2 | 0 | 0 | 86 | |

Events were graded according to Common Terminology Criteria for Adverse Events (CTCAE) version 5.0. Data are in n (%). Events reported as of 16/11/24. Source data are provided as a Source Data file.

tumors (target lesions) at day 36 respectively. A reduction or no change in tumor diameter was observed in 5/7 (71%) and 6/12 (50%) injected and non-injected tumors (target lesions) at day 92 respectively.

Analysis of OS according to dose/cohort revealed a median OS of 85, 122, 280, and 202.5 days for cohort 1, 2, 3, and 4 respectively. No significant difference between individual cohorts was observed when using Mantell-Cox Logrank test (Supplementary Fig. 1c). When groups were compared according to cohorts 1 and 2 vs 3 and 4 (lower vs higher), median OS was 107.5 days and 280 days respectively, however no statistical significance ($P = 0.1726$) was achieved (Supplementary Fig. 1d). The non-significant trend observed in individual cohort analysis and grouped cohorts suggests longer OS when using the dose of either cohort 3 or 4. Comparison of platinum resistant and refractory patients revealed a median OS of 138 and 235 days ($P = 0.8696$) and a PFS of 100.5 and 98 days respectively ($P = 0.7377$) (Supplementary Fig. 1e).

Longer overall survival was observed in 3 patients (302-06, 301-02, and 301-05) for approximately 12 months after the start of treatment. Patient 302-06 (cohort 1) with platinum resistant low-grade serous ovarian cancer (LGSOC) presented with 12 lines of previous cancer treatments, including three lines of hormonal therapy. The patient completed the trial and was enrolled into the extension period for eight months, after which the patient eventually died, 306 days after enrollment.

Patient 301-02 (cohort 2) with platinum resistant HGSOC was enrolled onto the trial 13 months after initial diagnosis, presenting with four lines of previous cancer therapy, consisting of two lines of chemotherapy, one surgical intervention and one line of targeted therapy (bevacizumab). The patient completed the trial and was evaluated as progressive disease at day 92. The patient came off the trial on day 92 and started gemcitabine and topotecan to treat metastatic disease on days 128 and 247 respectively. Altogether, the patient survived 434 days after enrolment.

Patient 301-05 (cohort 3) with platinum resistant HGSOC was enrolled onto the trial 15 months after initial diagnosis, presenting with one line of chemotherapy and one surgical intervention. The patient had amongst the highest number ($n = 4$) of target lesions at the start of the trial and three of these lesions made up 50% of all non-injected lesions in the trial that showed a reduction in tumor diameter as measured by CT (Supplementary Fig. 1b). This included a 35.8% reduction in a tumor diameter from baseline in the left lung at day 36, as well as a 69.2% reduction in a lesion located in the pelvic region at day 92. The patient completed the trial with stable disease at day 92 and entered the extension period, during which the patient received three additional treatments of TILT-123 and pembrolizumab. The patient eventually discontinued and received mirvetuzimab soravtansine. The patient is alive at the time of data cutoff 463 days after trial enrolment.

TILT-123 in the ICI resistant setting was observed in patient 301-04 (cohort 2) with platinum resistant serious ovarian cancer, who previously received palliative pembrolizumab one year prior to enrolment. The patient showed a 36% reduction in injected tumor diameter at day 36, however the patient exhibited disease progression at the next visit and died 93 days after trial enrolment.

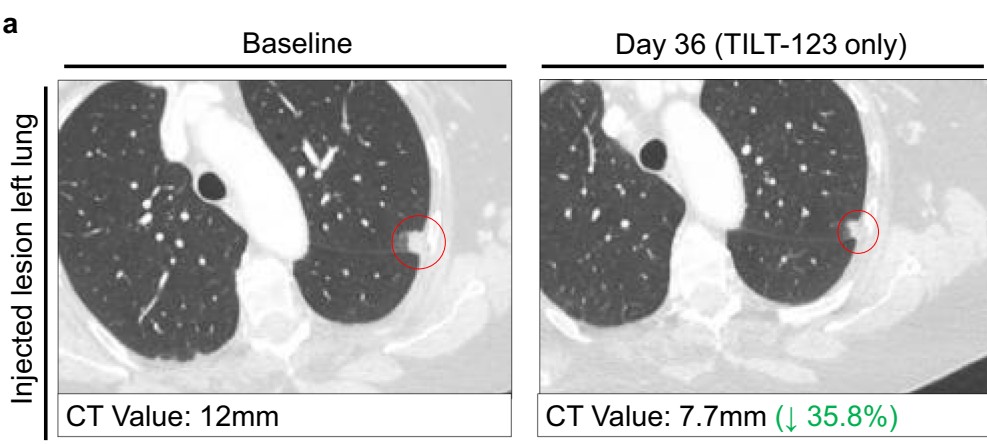

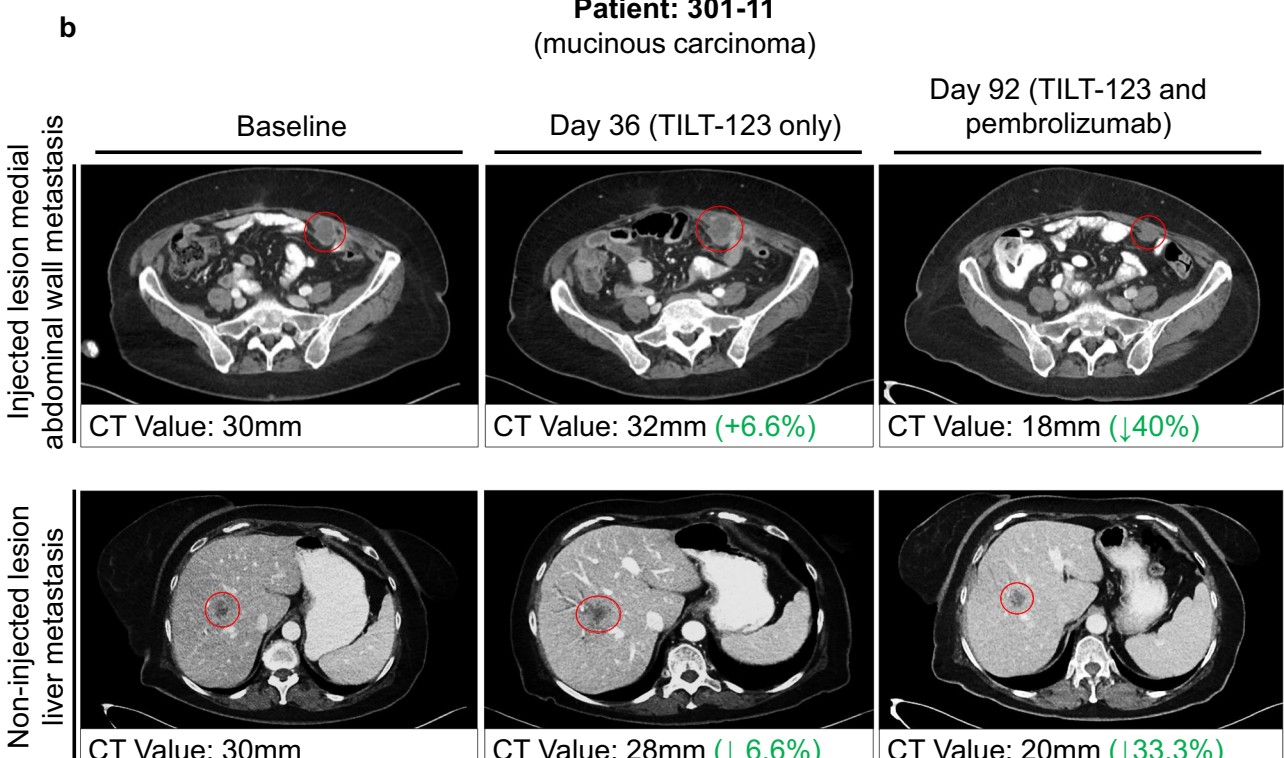

**Fig. 2 | Examples of treatment induced changes in tumour size. a** CT scan of a injected lesion (circled in red) in the left lung of patient 301-05, before (left) and after (right) treatment on day 36, indicating a 35.8% reduction in tumor diameter from 12 mm to 7.7 mm following three injections of TILT-123 only. **b** CT scan of injected and non-injected lesions in medial abdominal wall (top row) and liver metastasis (bottom row) of patient 301-11. CT scans display before treatment (left) and after treatment on day 36 (TILT-123 only) (middle) and day 92 (TILT-123 and pembrolizumab) (right). CT value change from baseline to day 92 of injected abdominal wall metastases indicates a 40% reduction in tumor diameter from 30 mm to 18 mm, whilst the non-injected liver metastasis shows a 33.3% reduction in tumor diameter from 30 mm to 20 mm.

A partial response was achieved in a patient with mucinous adenocarcinoma (301-11, cohort 4), a subtype notoriously unresponsive to chemotherapy. The patient was enrolled onto the trial, refractory to two prior lines of therapy (progression on first-line paclitaxel-carboplatin, then second line oxaliplatin-capecitabine). The patient presented with three target lesions, one injected tumor located on the medial abdominal wall and two non-injected tumors located on the lateral abdominal wall and liver (Fig. 2b). The patient responded well to TILT-123 and pembrolizumab, reaching the primary endpoint with an overall 30% reduction in target tumor sum. The patient is alive at data cutoff and enrolled onto the extension treatment period. Fever (once with grade 3 delirium) was reported after i.t. dosing of TILT-123 on day 22.

## Pharmacokinetics
One hundred ten blood samples were analysed from nine patients at seven different days for detection of TILT-123 specific DNA by qPCR (Fig. 3a). On each day, three sampling time points at 0 h, 1 h, and 16 h were included to evaluate acute changes in viral pharmacokinetics.

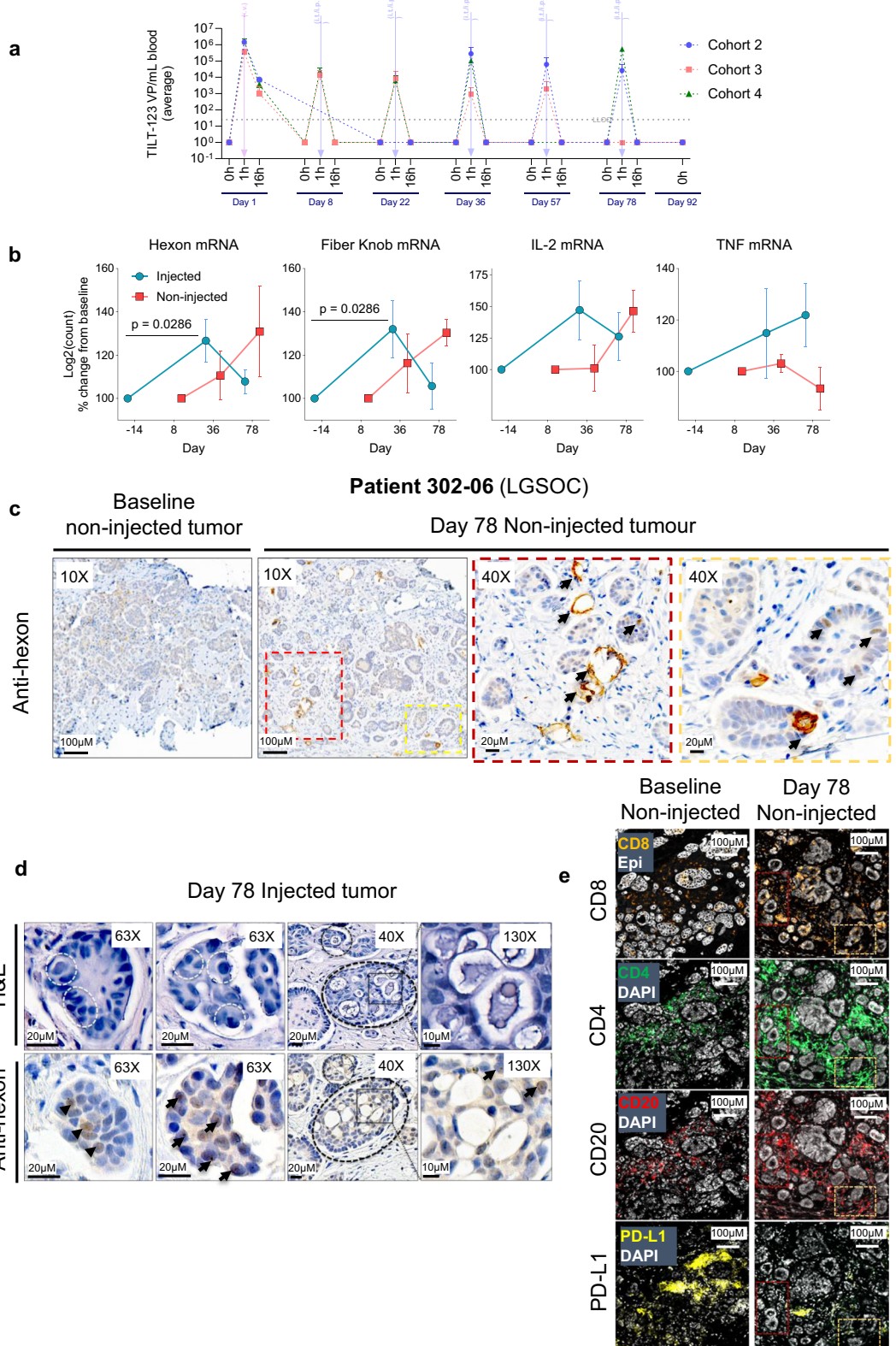

**Patient 302-06** (LGSOC)

TILT-123 was detected in 100% of samples 1 h after injection and reached maximum detected concentration (vp/ml) after the first i.v. injection. TILT-123 was undetectable in most samples 16 h after injection, except for samples collected 16 h after the first intravenous injection. No clear association between cohort and concentration detected in the blood was observed, although the highest concentration ($10^6$ vp/ml) of virus detected was in patient 302-10 (cohort 4) 1 h

after i.v. injection. The highest concentration of virus detected in the blood after i.t. injection at any time point was from patient 301-03 (cohort 2) on day 36 ($10^6$ vp/ml).

TILT-123-specific viral DNA was not detected in any of the urine, feces or saliva samples analysed (Supplementary Table 5).

Twenty seven tumor biopsies from 5 patients were analysed for expression of mRNA specific to adenovirus structural genes and TILT-

**Fig. 3 | Pharmacokinetics, virus detection and immunomodulation of tumors.**
**a** Concentration of TILT-123 present in whole blood of patients in cohorts 2–4 during the trial, as determined by qPCR ($n = 7$). **b** Detection of TILT-123 in injected and non-injected tumors as determined by quantification of adenovirus hexon and fiber knob mRNA and TILT-123 transgenes IL2 and TNF ($n = 6$). **c** Detection of TILT-123 in non-injected tumors of patient 302-06 at baseline compared to day 78, as determined by anti-hexon immunohistochemistry, where areas of interest are boxed in yellow and red and black arrows indicate hexon positive (brown DAB stain) adenovirus (TILT-123) positivity. **d** Characterisation of an injected tumor from patient 302-06 showing basophilic inclusions in white dotted circles (top row) which co-localise with hexon (brown DAB stain) positive stain indicated by black

arrows (lower row). Right quadrant shows microcalcifcations within tumor nests circled in black dotted circles which co-localise with hexon positivity indicated by black arrows. Magnified area of interest is shown on the right column of the quadrant. **e** Multiplex immunofluorescence of the matching biopsies above in figure **c**, indicating changes in CD8+, CD4+, CD20+ and PD-L1+ expressing cells from baseline compared to day 78. Matching areas of interest boxed in yellow and red. Data sets in a and b are presented as mean ± SEM and significance represented by exact *p*-values. Comparisons was evaluated two-tailed Mann Whitney *U* test. n number of patients (biological replicates). Source data are provided as a Source Data file.

123 cytokine transgenes. Significant adenovirus hexon ($P = 0.0286$) and fiber knob ($P = 0.0286$) mRNA expression was detected in injected tumors at day 36 compared to baseline (14 days after the previous treatment). An increase in hexon ($P = 0.1650$) and fiber knob ($P = 0.0781$) mRNA was detected in non-injected tumors at day 78 compared to baseline (Fig. 3b). Additionally, changes in expression of TILT-123 associated cytokine transgenes were detected. IL-2 increased in injected tumors at day 36 ($P = 0.1177$) and non-injected tumors at day 78 ($P = 0.1712$), whilst TNF increased in injected tumors at day 78 ($P = 0.1369$) and non-injected at day 36 ($P = 0.5066$). Interestingly there was a drop in IL-2, hexon and fiber knob expression in the injected lesions from day 36 to 78, whilst expression increased in the non-injected lesions at the same time points. This dynamic likely indicates systemic dissemination of TILT-123 from injected lesions to non-injected tumors.

Preliminary proteomic analysis of adenovirus hexon protein by immunohistochemistry complemented the qPCR and mRNA analysis. Adenovirus hexon positive cancer cells (Epi+) were detected in both injected and non-injected tumor biopsies from patient 302-06 on day 78 (21 days after the previous treatment) (Fig. 3c, d). These virus positive regions also co-localised to areas of high immune cell infiltration, represented by an increase in CD8+, CD4+ and CD20+, as well as a decrease in PD-L1+ immune clusters (Fig. 3e).

### Immunomodulation of tumors

Nineteen tumor biopsies and 21 non-injected tumor biopsies were collected from 6 patients at 4 time points from cohorts 1–3. Out of these 40 biopsies, 34 samples passed quality control and were analysed by immunohistochemistry and multiplex immunofluorescence.

Prominent histological features were observed in the injected and non-injected tumor biopsies from patient 302-06 on day 78, which included presence of adenovirus hexon positive stained epithelial cells which were not present at baseline (Fig. 3c, d). Additionally basophilic inclusions, psammoma bodies, microcalcifications, and inflammation were observed co-localised to adenovirus hexon positive cells in Epi+ tumor nests within the injected abdominal wall tumor biopsy (Fig. 3d). Both injected and non-injected tumors were infiltrated with both CD4+ and CD8+ T cells as well as B cells (Fig. 3e).

Quantification of multiplex immunofluorescence from tumor biopsies revealed immune profiles that potentially rationalise combined use of TILT-123 and pembrolizumab, whilst also revealing immunomodulation profiles associated with response. Analysis ($n = 6$) of intratumoral CD8 T cells and CD56 immune cells showed a significant increase in percentage of CD8+, PD-1+, CD45+ cells ($P = 0.0470$) and CD56+, PD-1+, CD45+ ($P = 0.0229$) in the injected lesions post-treatment when compared to baseline (Fig. 4a). There was a non-significant trend increase in both populations in the non-injected tumors. Individual fold changes were varied between patients in both injected and non-injected tumors. A high fold change increase in PD-1 expressing CD8 T cells and NK cells was observed in patient 301-03 following treatment in both injected (59 and 34 fold increase at day 78 respectively) and non-injected tumors (14 and 71 fold increase at day 78 respectively) (Fig. 4a). A fold change increase from baseline in

PD-1 expressing CD8 T cells was also observed in patient 302-07 (11-fold), 302-09 (19-fold) and 302-06 (twofold). Analysis of CD4 T cells and B cells revealed a significant increase in percentage of intratumoral CD4+, CD45+ cells ($P = 0.0165$) in injected tumors at day 36, and a non-significant increase in CD20+, CD45+ cells ($P = 0.0947$) in injected tumors at day 78 when compared to baseline (Fig. 4a). A fold change increase in CD4 T cells in either injected or non-injected tumors was observed in patient 302-09, 302-07, 302-06, and 301-03. Whilst a fold change increase in B cells in either injected or non-injected tumors was observed in patient 302-07, 302-09, 301-03, 302-06, and 301-02 (Fig. 4a).

### Humoral immunity is associated with treatment efficacy

One hundred and five serum samples from 15 patients at seven time points were analysed for detection of neutralizing antibodies (nAbs) against adenovirus. 7/15 (47%) of patients had detectable nAb titers against adenovirus at baseline, including patient 301-04 and 302-10 with low titers (1:64), 301-05, 302-06, and 302-09 with medium titers (1:256–1:1024) and 301-02 and 301-12 with high titers (1:4096). All patients nAb titers increased following treatment, with no clear association between cohort and nAb titer developed (Fig. 4b).

Interestingly, a higher nAb titer (interpreted as strength of humoral response) following treatment with TILT-123 was significantly ($P = 0.00330$) associated with better imaging response (stable disease or partial response vs progressive disease) (Fig. 4c). Presence of nAbs at baseline did not correlate with imaging response (Fig. 4d). A higher nAb titer following treatment was also significantly ($P = 0.0109$) associated with longer overall survival (Fig. 4e) whilst presence of nAbs at baseline also significantly ($P = 0.0289$) associated with longer overall survival (Fig. 4f).

## Discussion

Here we report the results of a single-arm, multicentre phase I dose escalation clinical trial evaluating safety and efficacy of hybrid i.v. and i.t. delivery of TILT-123, with systemic pembrolizumab for patients with extensively pre-treated platinum-resistant or refractory ovarian cancer. The trial confirmed that the treatment approach is safe and able to reach and elicit a significant immune response in both injected and non-injected lesions, despite the difficulty of the patient population. Anti-tumor activity and tendency for long survival were observed in several patients that had failed multiple lines of therapy and presented with a large tumor burden. This included patient 302-06 who survived 306 days after enrolment following 12 lines of failed therapies and patient 301-05 who is alive 463 days after enrolment presented with four large lesions.

Contemporary ovarian cancer trials regularly exclude platinum refractory patients because they are considered too challenging. This subpopulation is nevertheless included here, and in fact had longer PFS and OS than the resistant population. This suggests TILT-123 + pembrolizumab could be an attractive option for further study in these patients with dire unmet medical need.

Phase 1a patients have typically been extensively treated with several therapies prior to trial entry, rendering their tumors capable

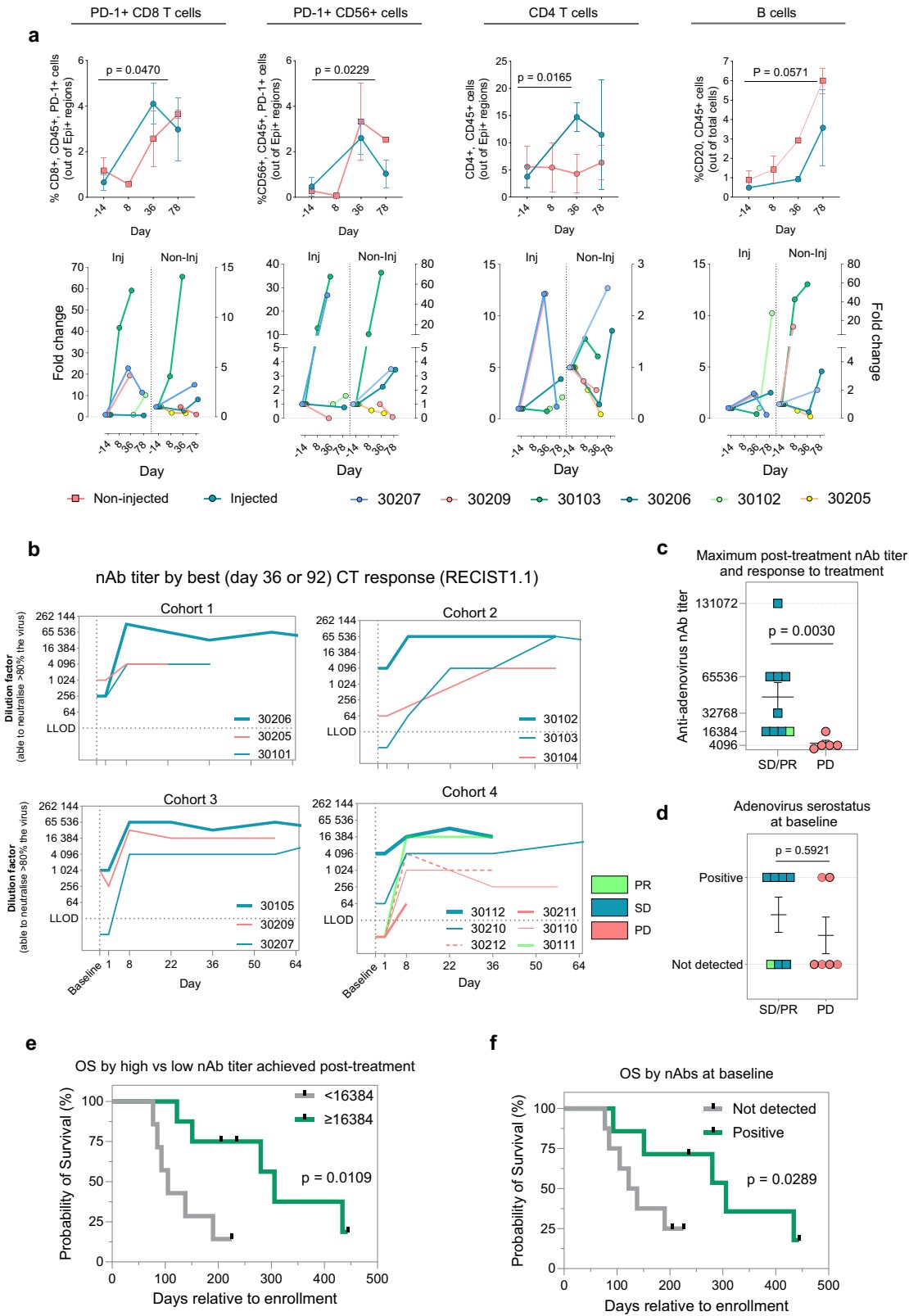

**a**

**b** nAb titer by best (day 36 or 92) CT response (RECIST1.1)

**c** Maximum post-treatment nAb titer and response to treatment

**d** Adenovirus serostatus at baseline

**e** OS by high vs low nAb titer achieved post-treatment

**f** OS by nAbs at baseline

of thwarting any therapeutic approach. It is therefore unusual to see high anti-tumor activity in these patients. The efficacy seen in this trial (64% disease control) compares favorably with phase 1a results obtained with mirvetuximab soravtansine (56% disease control), which was later FDA approved (following trials in earlier line healthier patients) for treatment of folate receptor positive platinum resistant ovarian cancer[23]. With that said, the sample size in the present dose

escalation study is too limited (total $n = 15$; $n = 5$ imaged patients at the highest dose) to assess efficacy reliably. Therefore, a larger cohort is required to be able to adequately evaluate response with sufficient statistical power. Additionally, the patients reported in this study may not be representative of the average patient population, for example patient 302-06 who was still alive 23 years after initial diagnosis.

**Fig. 4 | Immunomodulation of tumors and neutralizing antibodies as a potential biomarker. a** Percentage (upper row) and fold change (lower row) during treatment in injected and non-injected tumors for intraepithelial PD-1 expressing CD8 T cells, intraepithelial PD-1 expressing CD56 NK cells, intraepithelial CD4 T cells and all CD20 B cells present in biopsy (n = 6). **b** Neutralizing antibody titers generation across trial, stratified by cohort and best clinical response (RECIST1.1) with red indicating progressive disease (PD) and dark green indicating stable disease (SD) and light green indicating partial response (PR). **c** Association of best clinical response (RECIST1.1) and anti-adenovirus nAb titer developed following treatment with TILT-123 (n = 14). **d** Association of best clinical response and presence of nAbs at baseline line (**e**). Association of overall survival with anti-adenovirus nAb titer developed following treatment (n = 14). **f** Association of overall survival with presence of nAbs at baseline. Percentage is presented as mean ± SEM and significance represented as *$p < 0.05$, **$p < 0.001$. Comparison between groups was evaluated by two-tailed Mann Whitney $U$ test or Welch's $t$ test. Overall survival in **e** was evaluated using Log-rank (Mantel-Cox) test and Mann Whitney $U$ test for overall survival in **f**. n number of patients (biological replicates). Source data are provided as a Source Data file.

During the trial, three patients received i.p. dosing of TILT-123, all of which were well-tolerated. Notably, on day 36, the patient exhibited disease control, suggesting good tolerability, and supporting the feasibility of i.p. injection. Overall, i.t. and i.p. injections were well-tolerated by patients and although requiring some skill, local delivery of oncolytic virus was performed successfully without long-term implications to patient well-being. Virus particles were detected in both injected and non-injected lesions by mRNA and immunohistochemistry. Notably, actively replicating virus was observed in an immune infiltrated, non-injected tumor (302-06, day 78) 21 days after the last treatment with TILT-123. This suggests that TILT-123 was able to reach non-injected tumors, evade anti-viral immunity and maintain anti-tumor activity for a significantly prolonged duration. The patient was receiving amlodipine, also taken by 301-05 and 301-03, who both had high circulating virus titers many days after the last given treatment. Calcium blockers have been shown to increase blood virus titers of oncolytic adenoviruses such as ICOVIR[24].

Analysis of the PD-1 immune cell axis during the study revealed a significant increase in PD-1 expressing CD8+ T cells and CD56+ NK cells in injected tumors prior to the first infusion of pembrolizumab, providing a rationale for the combination approach. Indeed, these findings support the 'prime and boost' treatment design established in preclinical studies with TILT-123[11]. These data suggest that TILT-123 could sensitize ovarian cancer tumors to immune checkpoint blockade.

Presence of serum nAbs against adenovirus at baseline did not prevent systemic dissemination of TILT-123 to non-injected lesions, as shown by virus detection and changes to the tumor immune microenvironment. Partial data from the d8 biopsy time point of PROTA has been reported in a cross-trial analysis of d8 biopsies, demonstrating the presence of virus in non-injected tumours after the first intravenous dose of TILT-123[25]. In contrast, longer survival was observed in patients with nAbs against adenovirus at baseline, compared to those without detectable levels. Interestingly, a higher nAb titer following treatment was associated with clinical response and longer OS. Anti-adenovirus nAbs are traditionally thought to impair adenovirus transduction efficiency, but this could chiefly apply to small amounts of virus being released from infected epithelia into blood, in contrast to the large amounts of virus present in blood in this trial, which may overwhelm the opsonization capacity of nAbs. Additionally, interpretation of the nAbs data should be regarded with caution as multiple factors can influence overall survival, including overall health and age which play a role in the status of humoral immunity.

Notably, production of high anti-adenovirus nAbs titers in response to treatment may indicate systemic fitness of the humoral response. Our findings are accord with observations in a recent phase I trial with an oncolytic herpes virus for the treatment of glioblastoma[26]. Whether functionality of circulating antibody-producing cells reflects functionality in tumors in the context of this study, has yet to be elucidated. Indeed, augmentation of intratumoral B cells was observed, with a higher percentage of CD20+ B cells in injected tumors post treatment. We also noted presence of tertiary lymphoid structures in long term responders (302-06 and 302-07) at baseline, which appeared to mature during treatment. There is mounting evidence that tumors enriched in B cells and tertiary lymphoid structures predict response to immunotherapy in several types of cancers, including HGSOC[27–30]. A more defined role of antibody producing cells in HGSOC has been described, including production of autoantibodies directed to MMP14 overexpressed on tumor cells, and the role of antigen-specific and antigen-independent polyclonal IgA in directing the intratumoral immune response[31,32]. Interestingly, both studies speculate Fc mediated effector function as an opportunity for therapeutic targeting. We did observe an increase in intratumoral Fc effector cells, mainly CD56+ immune cells (NK cells), following treatment. In conclusion, the present study reports that treatment of platinum resistant or refractory ovarian cancer with TILT-123 and pembrolizumab is safe, well tolerated at all tested doses and can elicit prolonged disease control in some patients, despite extensive prior therapies. Hybrid i.v. and i.t./i.p. delivery of TILT-123 enables immunomodulation and anti-tumor responses in both injected and non-injected lesions. Translational analyses indicated a potential role of the humoral response as the first identified biomarker for treatment with TILT-123 and pembrolizumab.

We are currently expanding the current study into a phase 1b using TILT-123 and pembrolizumab in combination with PEGylated liposomal doxorubicin. The phase 1b uses the virus and pembrolizumab dose from cohort 4. The final dose recommended for the next phase of trials evaluating efficacy has yet to be determined as the ongoing phase 1b is not completed yet.

## Methods

### Ethics statement

The study was approved in Finland by the National Committee on Medical Research Ethics (Tukija) under the study number TUKIJA-405-2021. The study was approved in USA by FDA under the IND# 027209. IRB approval number at Mayo Clinic - 22-000078. Informed consent was obtained from all patients reported in this trial.

The study design and conduct complied with all relevant regulations regarding the use of human study participants and was conducted in accordance with the criteria set by the Declaration of Helsinki.

### Patients and Methods

Fifteen patients were enrolled onto the trial (NCT05271318) as listed in Table 1. Key eligibility criteria included the requirement of adequate hepatic (total bilirubin: ≤1.5 × ULN and AST and ALT: ≤2.5 × ULN [≤5 × ULN for participants with liver metastases]) and renal function (GFR: >45 ml/min), and WHO/ECOG performance score of 0–1. Patients should have histologically confirmed ovarian cancer (including fallopian tube and primary peritoneal cancer) resistant or refractory to platinum-based chemotherapy and life expectancy longer than 3 months. At least one tumor >14 mm in diameter was required for i.t. injection. In patients with carcinomatosis not amendable to i.t. injection, i.p. injection was used.

Platinum refractory was defined as progression of cancer within 30 days of the most recent dose of cisplatin or carboplatin (in the absence of prior response or disease control). Platinum resistant was defined as progression of cancer within 183 days of the most recent dose of cisplatin or carboplatin.

Key exclusion criteria included active (past two years) autoimmune disease requiring systemic immunosuppressive drugs, treatment with any anti-cancer therapy within 30 days, prior treatment with an anti-PD-1, anti-PD-L1, or anti-PD-L2 agent or other agent directed towards T cell immune checkpoint receptor that resulted in discontinuation due to Grade 3 or higher irAE and known contraindications to pembrolizumab.

## Preparation of TILT-123 dosage

Prior to administration, TILT-123 was resuspended in 0.9% saline and administered in 1.0–5.0 mL volume for i.t. injections, 50–51 ml for i.p. injection and 10.0–40.0 mL for i.v. injections, depending on the dose cohort. For i.t. injections, up to 10 virus deposits in the tumors were performed. For i.p. injections a 22–26-gauge needle was used to inject virus directly into the peritoneal cavity. The patient was massaged for three minutes in four positions to allow dissemination of the virus in the peritoneal cavity.

## Design

To evaluate the safety (including DLTs) of combining TILT-123 with pembrolizumab, we conducted a 3 + 3 design dose escalation design[33–35]. Virus dose increased between cohorts 1–4 and not intrapatient. Patients showing possible beneficial treatment effects in the initial treatment period entered a treatment extension period (continuing the same respective dose as initial treatment period) until completion of up to 38 injections of TILT-123 and 35 cycles of pembrolizumab (up to approximately 2 years). Eligibility to continue treatment was assessed case-by-case by the sponsor and the clinical investigator.

Safety and efficacy were evaluated on day 92, unless clinical symptomatic progression leading to treatment discontinuation was observed prior to this day. A 12-month safety follow-up visit period was included (day 122, 3 months, 6 months, and 12 months after the end of the initial treatment period), as well as a 5-year GMO follow-up period.

DLT is defined to be a toxicity event that prevents further administration of the agent at that dose level. DLT is specifically defined as any death not clearly due to the underlying disease or extraneous causes, or as any treatment-related Grade ≥ 3 non-hematologic AE, or any Grade ≥ 4 hematologic AE according to the NCI-CTCAE v 5.0, occurring during the DLT window of observation period. The DLT window of observation will be from day 1 to day 57, covering from the first TILT-123 injection up to 21 days after the first dosage of TILT-123 in combination with pembrolizumab. The decision tree for the 3 + 3 dose escalation design and DLT implementation can be found in Supplementary Fig. 4a, b.

AE grades were used to define this AE endpoint, per Adverse Events by NCI-CTCAE version 5.0 and if not possible to capture by these criteria the investigator will use AE criteria described in Supplementary Fig. 4c.

## Treatment

In the initial treatment period, each patient received six injections of TILT-123 and three infusions of pembrolizumab. The first TILT-123 injection on day 1, was i.v. and ranged from $3 \times 10^{11}$ to $4 \times 10^{12}$ VPs. The following TILT-123 injections ranged from $1 \times 10^{11}$ to $5 \times 10^{11}$ VP and were delivered locally (i.p., and/or i.t.) on day 8, 22, 36, 57, and 78. Pembrolizumab (200 mg) was administered i.v. Q3W after TILT-123 injections on day 36, 57, and 78. DLT were defined as toxicities related to the study treatment that would prevent further administration of the agent at that dose level and would meet certain criteria. Pembrolizumab administration could be continued in the case patient needed to discontinue TILT-123. At least one tumor was injected during i.t. dosing, with the agent distributed evenly to multiple locations inside each injected tumor.

A graphical summary of the treatment regime can be found in Fig. 1d.

## Outcomes

The primary endpoint was safety based on types, frequency, and severity of Adverse Events (AE), including monitoring of vital signs, electrocardiogram (ECG), and safety laboratory results. Adverse events were graded according to Common Terminology Criteria for Adverse Events (CTCAE) version 5.0. A complete blood count (with differential), liver tests and kidney tests were conducted by routine laboratory testing. Assessment of adverse events (AEs) for the combination of TILT-123 + pembrolizumab was descriptive.

Secondary objectives included maximum tolerated dose (MTD), tumor response by standard imaging, changes in CA125 levels, progression free survival (PFS), overall survival (OS), immune response towards the virus (neutralizing antibodies) and tumor, virus persistence, and virus shedding. CA125 was used to calculate progression when imaging data was not available, as defined as doubling in value from the upper limit of normal or doubling from the nadir on 2 occasions at least 1 week apart.

Exploratory objectives included evaluation of changes in patients' tumors or metastases and assessment of biological effects in biopsy samples. Biological effects in biopsy samples included analysis of changes in proportions of different immune cells markers, expression of immune-related genes, as well as treatment related pathological changes.

Shedding into urine, saliva and feces was studied.

TTP was evaluated but was not an official secondary objective. More detailed definitions for MTD, PFS and TTP are described in Supplementary Fig. 4d.

## Assessment of treatment response

To determine the clinical response of TILT-123 and in combination with pembrolizumab, two CT imaging time points evaluating injected and non-injected lesions were included on day 36 (prior to pembrolizumab) and 92 and were assessed for response by both iRECIST and RECIST 1.1. Patients were enrolled into the extension period if benefit was observed by day 92.

OS and PFS data were retrieved from the electronic clinical trial records. Data cut off for OS, PFS, and TTP was 16/05/24.

For patient response analysis the following criteria were used to define DCR and ORR.

$$DCR = CR + PR + SD$$
$$ORR = CR + PR$$

## Analysis of virus persistence and shedding

The presence of the virus (virus persistence) was investigated from whole blood and shedding (urine, feces, and saliva) samples by qPCR using primers targeting an engineered region of the virus (IRES-IL-2) that is not naturally present in wild type adenovirus nor the human genome. The primers and probes were used at a concentration of 100 μm.

FWD 5′-CATGCTTTACATGTGTTTAGTCGAG-3′
RVS 5′-TAGTGCAATGCAAGACAGGAG-3′
PRB 5′-/56-FAM/TTGCATCCT/ZEN/GTACATGGTTGTGGC/3IABkFQ/-3′

## Serum Anti-adenovirus antibodies

Anti-adenovirus antibodies were measured by neutralizing antibody assay described in more detail previously[36]. In brief, A549 were seeded overnight in 96-well plates and the next day serial dilutions of complement inactivated serum samples from patients were incubated with Ad5/3-luciferase for 1 h. A549 cells were then infected with Ad5/3-luciferase pre-incubated with serum overnight and luciferase expression quantified the following day using Luciferase Assay System (Promega, WI, USA). The assay can be easily set up in any laboratory and takes 3 days assuming cells are kept in culture. The minimum and maximum dilutions included in the original assay set up were 1:64 and 1:262144. The titer was

determined as the lowest dilution of serum that blocked at least 80% of luciferase expression.

## IHC and multiplex immunofluorescence staining

To assess treatment related immune modulation and distribution, injected and non-injected tumors were biopsied at baseline, day 8, 36, and 78, when feasible. Biopsies were formalin fixed and paraffin embedded prior to sectioning and staining (including H&E). For assessing virus distribution to tumors by immunohistochemistry, anti-Adenovirus-Hexon antibody (Millipore, AB1056, 1:1000) was used.

For multiplex immunofluorescence (IF) primary antibodies included anti-CD56 (CM, 156R-94, 1:100, lot207206), anti-CD8 (Dako, M7103, 1:300, lot20071297), anti-PD-1 (LSBio LSB12784, 1:150, lot69182), anti-CD45 (Dako, M0701, 1:100, lotM0701), anti-Ecadherin (CST, 3195, 1:200, lot15), anti-pan Cytokeratin (Abcam, ab9377, 1:200, lot104586-14), anti-CD4 (Abcam, ab133616, 1:400, lot1075222-5), anti-CD20 (Thermo, MS-340, 1:200, lot340S2110A), anti-PD-L1 (CST, 13684, 1:200, lot18).

Stained slides were scanned on Zeiss Azio Scan Z.1 with 20× (NA 0.8) objective and quantified using Cell profiler version 4.2.5. Epi+ regions were designated tumor positive regions.

Multiplexed biopsy samples were quality checked and adequate regions of interest were identified. The first- and second round ROI-images were aligned based on nuclear 40, 6-diamidino-2-phenylindole signal. Pixel classification module of Ilastik was used to detect the epithelium/ stroma. Nuclei were segmented from 40, 6-diamidino-2-phenylindole channel using pretrained deep learning segmentation model. Analysis was processed using Cell profiler software. Cell regions were defined by dilating the nuclear segmentation. Mean channel intensity of individual markers co-localizing in each segmented cell was determined.

IHC and multiplex immunofluorescence was performed at the Institute for Molecular Medicine Finland (FIMM) Digital Microscopy and Molecular Pathology Unit supported by HiLIFE and Biocenter Finland.

H&E images were evaluated for possible treatment-induced changes by an experienced pathologist.

## Gene expression profiling

NanoString nCounter® gene expression analysis was performed on RNA samples from all tumor biopsies utilizing the nCounter® Digital Analyzer (NanoString Technologies, Seattle, USA). Gene expression was assessed with an nCounter® Human PanCancer. Immune Profiling Panel and additionally included a panel of adenovirus-associated genes: Ad5 Hexon, E1A and Ad3 Fiber knob. Genes considered to be differentially expressed between the groups of interest when $p$ value < 0.05. Data are presented as log2 (counts) normalized to baseline.

## Statistics and Reproducibility

Statistical analysis was performed using GraphPad Prism 9.4.1. For group analyses, two-tailed parametric t tests or nonparametric Mann–Whitney $U$-tests were used to compare groups. Mantell-Cox Logrank test was used to compare OS and PFS between groups including dose/cohorts, platinum status and post-treatment nAb titer. Two-tailed Wilcoxon matched-pairs t test was used to calculate significance for changes across time.

Sectioning, H&E and multiplex immunofluorescence staining, and analysis were repeated once due to limited biopsy material. Images in Fig. 4 represent observations of one patient (302-06).

## Role of funding source

TILT Biotherapeutics Oy was involved in the study design, data analysis and interpretation, writing and submission of the report for publication.

## Reporting summary

Further information on research design is available in the Nature Portfolio Reporting Summary linked to this article.

## Data availability

Source data are provided with this paper. Individual de-identified participant data related to adverse events will be shared in the source data file. A redacted version of the Study Protocol is provided in Supplementary files. Source data are provided with this paper.

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

## Acknowledgements

We thank Merck & Co., Inc. Rahway, NJ USA for their collaboration. We thank Minna Oksanen for expert assistance. This study was supported by Jane and Aatos Erkko Foundation, EU Horizon Grant 811693 (UNLEASHAD) (A.H.), Finnish Cultural Foundation (J.H.A.C, D.C.A.Q., E.J., S.B.), EU Horizon 2020 Research and Innovation Programme under the Marie Skłodowska-Curie Grant agreements (No 813453) (A.H., J.H.A.C.), TILT Biotherapeutics Ltd, HUCH Research Funds (VTR) (A.H.), Cancer Foundation Finland (A.H., J.H.A.C., T.K.), Sigrid Juselius Foundation (A.H.), K. Albin Johanssons Foundation (J.H.A.C, T.K., S.B.), Selma and Maja-Lisa Selander's Fund in Research in Odontology (Minverva Foundation) (J.H.A.C), Ida Montinin Foundation (J.H.A.C, T.K., S.B.) and the Finnish Red Cross Blood Service (A.H.). We thank Albert Ehrnrooth and Karl Fazer for research support. We thank the Department of Defense (DOD) for supporting this study (OC220391) (A.H.). We are grateful for the valuable contributions of the FIMM Digital Microscopy and Molecular Pathology Unit supported by HiLIFE and Biocenter Finland for immuno-histochemistry services. We are grateful to Annabrita Schoonenberg, Katja Välimäki and Teijo Pellinen for their technical expertise and invaluable contributions. TILT Biotherapeutics funded the study design, data collection and analysis.

## Author contributions

M.B., J.H.A.C, J.M., S.P., D.C.A.Q., T.K., E.J., L.H., S.S., R.H., C.K., A.K., V.C.C., J.M.S., and Ak.H: Conceptualization, design, investigation, methodology, acquisition, data curation, analysis, interpretation, visualization, drafting, and editing. M.V.H., V.A., S.B., N.O., T.P., An.H., K.V., and A.P.: acquisition, data curation, analysis, and editing. S.Z., S.G.V.K., and S.R.: Administrative, technical, or material support. T.A., D.A., S.R., J.S., J.K., J.C., M.J.C., and J.G.: Conceptualization, design, investigation, methodology, acquisition, interpretation, drafting and editing.

## Competing interests

A.H. is shareholder in Circio Holdings ASA. A.H., A.K., C.K., J.H.A.C., J.S., D.C.A.Q., S.S., R.H., L.H., and V.C.C. are employees and shareholders in TILT Biotherapeutics Oy. MJC is an employee and shareholder in Merck & Co., Inc. M.S.B. is on the TILT Biotherapeutics Scientific Advisory Board. Other authors declare no conflicts of interest.

## Additional information

**Matthew Stephen Block** [1,10], **James Hugo Armstrong Clubb**[2,3,10], **Johanna Mäenpää**[4,5], **Santeri Pakola** [3], **Dafne Carolina Alves Quixabeira**[2,3], **Tatiana Kudling**[2,3], **Elise Jirovec** [2,3], **Lyna Haybout** [2,3], **Mirte van der Heijden**[3], **Sanae Zahraoui**[2], **Susanna Grönberg-Vähä-Koskela**[3], **Sini Raatikainen**[2], **Victor Arias**[3], **Saru Basnet**[3], **Nea Ojala** [3], **Teijo Pellinen** [6], **Annabrita Hemmes**[6], **Katja Välimäki** [6], **Annukka Pasanen**[7], **Tuomo Alanko** [4], **Daniel Adamo** [1], **Susan Ramadan**[4], **Jorma Sormunen** [4], **Juha Kononen**[4], **Julia Wanda Cohen**[8], **Michael Jon Chisamore** [8], **John Goldfinch**[2], **Suvi Sorsa**[2,3], **Riikka Havunen** [2,3], **Claudia Kistler**[2], **Aino Kalervo**[2], **Víctor Cervera-Carrascon**[2,3], **João Manuel dos Santos**[2,3] **& Akseli Hemminki** [2,3,9] ✉

[1]Mayo Clinic Cancer Center, Rochester, MN, USA. [2]TILT Biotherapeutics Ltd, Helsinki, Finland. [3]Cancer Gene Therapy Group, Translational Immunology Research Program, University of Helsinki, Helsinki, Finland. [4]Docrates Cancer Center, Helsinki, Finland. [5]Tampere University, Faculty of Medicine and Medical Technology, Tampere, Finland. [6]Digital Microscopy and Molecular Pathology Unit, Institute for Molecular Medicine Finland, Helsinki, Finland. [7]Department of Pathology, University of Helsinki and Helsinki University Hospital, Helsinki, Finland. [8]Merck & Co. Inc., Rahway, NJ, USA. [9]Comprehensive Cancer Center, Helsinki University Hospital, Helsinki, Finland. [10]These authors contributed equally: Matthew Stephen Block, James Hugo Armstrong Clubb. ✉e-mail: akseli.hemminki@helsinki.fi

