## [Transparent Peer Review file · Nature Communications]

The oncolytic adenovirus TILT-123 with pembrolizumab in platinum resistant or refractory ovarian cancer: the phase 1a PROTA trial

Corresponding Author: Professor Akseli Hemminki

Version 0:

Reviewer comments:

Reviewer #1

(Remarks to the Author)

Summary

This innovative early phase trial investigates a new, combinatorial approach to make immunologically 'cold' tumors 'hot'. The authors' method involves an oncolytic adenovirus, previously investigated in early phase clinical trials (ref 1 - Pakola SA, Peltola KJ, Clubb JHA, Jirovec E, Haybout L, Kudling TV, et al. Safety, efficacy, and biological data of T cell-enabling oncolytic adenovirus TILT-123 in advanced solid cancers from the TUNIMO monotherapy phase I trial. Clin Cancer Res. 2024), and anti-PD-1 checkpoint inhibitor pembrolizumab, which is in widespread clinical use. This is the first time these treatments have been used in combination.

There are several noteworthy findings from this study. The experimental intervention appeared safe and tolerable, with most significant toxicity limited to lymphopenia. Survival outcomes were not inferior (but also not superior) to those expected of pretreated, platinum resistance/refractory HGSC; particularly welcome was the finding of a partial response in a patient with mucinous carcinoma of the ovary for which there are limited therapeutic options. The combination approach appears to have elicited responses in both injected and non-injected tumors. There was an early finding of a putative biomarker in serum anti-adenovirus neutralizing antibodies (nAbs), an encouraging result with future studies in mind. This study, therefore, is original and would be of great interest to those working in gynecologic malignancies and cell therapies.

COMMENTS:

1) We note that the authors used CA-125 as a secondary endpoint and as a metric of progression when imaging was not available. To interpret the relevance of this as a metric, it would be useful to know whether all patients had elevated CA-125 at baseline and if the study adhered to established GCIG criteria for CA125 response. Only this standard GCIG criteria should be used to assess CA125 response. CA125 response is not a clinically meaningful end point in this recurrent disease setting. "CA125 levels decreased below baseline levels in 47 60% of the patients at least at one time point" - this should be removed as not a validated end point

2) The exploratory objective, "evaluation of changes in patients' tumors or metastases and assessment of biological effects in biopsy samples" would be strengthened by some detail regarding what changes or biological effects are being referenced here.

3) Intraperitoneal dosing of the experimental agent is mentioned in the introduction and methods but then is not referred to again in results or discussion, with mentions limited to intratumoral injection. Please clarify if any patients receive intraperitoneal injection of the agent?

4) Did any patients suffer adverse events as a direct result of injection into the tumor (for example, the patient with hemoperitoneum grade 3) as distinct from systemic effects of the treatment? It may be of benefit to clarify this and also to report if there were any challenges in local delivery of the oncolytic virus.

5) How was platinum resistance and refractory defined?

6) Throughout the results section would try to avoid including conclusions and report only the results.

7) The median OS is 27 weeks and this would be within the expected survival range for this patient population. Would therefore consider omitting this "Notably, more than half of the patients survived longer than 6 months, which is encouraging in a highly pre-treated population with advanced platinum resistant or refractory ovarian cancer." as the findings are not notable and if included I would report 95% confidence intervals to clarify

8) Line 62 "The quantification of neutralizing antibodies against adenovirus provides a preliminary measure to predict the efficacy of treatment. Investigation of TILT-123, pembrolizumab"

It is not clear how clinicians would apply this in clinical practice and why RECIST response would not instead be used to assess response. What is the turn around time for this assay and would it be readily available?

9) Line 276 "The overall response rate (ORR) was 7.1% (1/14) and 20% (1/15) at the highest dose level." Is there a typo should it be 1/5 to equate to 20%.

10) The methods describe lesion disease control rate but in the results "Disease control rate" is reported so it needs to be clarified if they are referring to RECIST "SD and CR/PR for targets and non targets" or are they referring to the response in the injected lesions. This should be more clearly defined in the methods and the results- was this a pre-specified secondary endpoint?

11) Line 311 - "The patient came off the trial but surprisingly survived for 434 days from enrolment. Such cases are compatible with pseudoprogression resulting in treatment interruption despite anti-tumour benefits" Did the patient receive any subsequent therapy? If so then this sentence must be reworded as it is misleading and any OS benefit could be attributed to a different anticancer intervention

Reviewer #2

(Remarks to the Author)

In this study, the authors present data from a phase I trial of an oncolytic adenovirus Ad5/3 E2F-D24-hTNF α -IRES-hIL-2 with pembrolizumab in patients with recurrent ovarian cancer. The vector is a chimeric Ad5/3 backbone with a 24bp deletion in the E1A CR2 region and E1 expression driven by an E2F promoter. The vector also encodes two transgenes, TNF α and IL-2. The first injection was given IV, with subsequent injections given IT. Pembrolizumab was added from day 36 to allow assessment of virus-only activity and safety. There were 4 dose cohorts of increasing viral dose; pembrolizumab dose was constant. There were only two formal assessment points, days 36 and 92.

The headline data are that treatment was safe, with no DLT. The TEAE are as one would expect from prior trials of oncolytic adenovirus and also pembrolizumab with no suggestion of increased pembrolizumab toxicity. There was no cytokine release syndrome but an interesting drop in lymphocyte counts after the first virus dose, and subsequent oscillations in both lymphocyte and neutrophil counts.

There was one partial response in a patient with mucinous carcinoma. In all other patients, best response (by RECIST) was SD or PD. Three patients, including the one who had had a partial response, were alive at data cutoff.

There was extensive translational research, with multiple biopsies and blood samples, for which the authors (and more importantly the participants) should be congratulated. Key points were possible detection of Hexon and Fiber knob mRNA in uninjected lesions, suggesting infection of those lesions. There were also some suggestions of immune cell infiltration following injections, although the true meaning of these changes is not clear.

Overall, this type of trial is highly challenging, and the authors have undertaken a huge amount of work to complete the study. However, the participant number is low at 15, making it hard to reach definitive conclusions.

Major points:

1. Patients and methods - Table 1 and patient characteristics. Critical features for these patients, including BRCA1/2 mutation status, MMR status, are required. For the patients labelled as 'serous' (301-01, 301-03, 301-04), it is important to identify whether they had high or low grade serous disease – simple IHC should be able to answer that question. This is of relevance because LGSOC often grows very slowly with very long periods of stable disease and long overall survival (patient 302-06 being a very clear example).
2. Assessment of treatment response. The authors use unusual definitions of DCR and ORR – 'lesion' control rate and objective response rate. Are the authors excluding non-injected or non-target lesions here? The wording in the results text should ORR by RECIST1.1 and by iRECIST using the standard definitions (as shown in Table 1). Strictly speaking, DCR is not a RECIST term, so the authors should state clearly how they defined this (I think they used all patients who achieved PR and SD as best RECIST1.1 response)
3. Figure 1b and c should be presented as single lines – ie PFS and OS without the 'disease control' and 'no disease control' lines. This is particularly true for PFS – by definition, patients who have no disease control on day 92 will have a poorer PFS than those who have achieved disease control on day 92. A spider plot of individual patient data over time would also be helpful.
4. Similarly, Figures 1e-h should be presented as a single waterfall plot showing overall change compared to baseline for the whole cohort. It might be possible to present data from day 36 and day 92 separately in the supplementary data.
5. The efficacy section states that a 'decrease in CA125 levels was observed in 60% of patients at least at one time point'.

This is not a meaningful measure. What was the CA125 response rate by GCIg criteria? From the data presented in Figure 1d, I think that the CA125 response rate was 0%. This may not matter given the inflammatory nature of adenovirus infection, but the authors should use standard measurements.

6. Similarly, on lines 286-287, the authors state 'Evaluation of non-target lesions revealed only 1 out of 18 (6%) (Patient 301-10) progressing on day 36 and 92'. RECIST evaluation is a sum of all lesions (target and non-target), so discussing non-target lesions is not terribly helpful or meaningful.

7. The authors also discuss 'pseudoprogression'. This term requires lesions to get bigger since baseline but then to reduce in size. If a lesion reaches the RECIST definition of PD and then stays at the same size on the subsequent scan, that is iUPD rather than pseudoprogression. However, given that there was only 1 partial response and only two assessment points after the start of treatment, I am not sure how Supplementary Figure 1f can be plotted to show Dose Pseudoprogression.

8. Figure 3b. The possible (ie non-significant) demonstration of hexon and fiber in non-injections lesions is of great interest. However, what is the lower limit of detection here? In Figure 3a, the data are presented as VP/ml with LLOQ between 101 and 102. However, in 3b, the y-axis units are 'log2(Counts)'. It would be hard to imagine that there would be any detectable Hexon or Fiber knob mRNA or indeed IL2 and TNF using vector-specific primers on day -14 in any lesion, injected or uninjected.

9. Figure 3d – the white dotted arrows are supposed to show cytopathic effect, which is very far from obvious. Similarly, the black arrows are supposed to present viral inclusion bodies, which are again far from clear. Several of the black arrows point to interstitial space. There is also discussion of Mallory bodies, which I believe are hallmarks of liver disease, especially of alcoholic origin. Was a formal pathology review performed? And was the pathologist blinded to whether the sample was injected, uninjected, pre- or post-treatment?

10. Figure 3e is reported to show co-localisation of virus positive regions with areas of high immune cell infiltration, represented by an increase in CD8+, CD4+ and 372 CD20+, as well as a decrease in PD-L1+ immune clusters. There is no demonstration of virus positivity in Figure 3e and there is no formal analysis to correlate CD8 etc cell density with Hexon staining. However, I do note the finding of an increase in CD8 and NK cells in Figure 4a.

11. The humoral response data are very interesting, in particular the finding that those with PD had lower maximum post-treatment nAb titers, suggesting that those patients who cannot mount an Ab response to large virus doses have a poorer outcome. In terms of the KM curves in Figures 4a and f, these need to be regarded cautiously given the multiple factors that can determine overall survival, including age, prior treatments, debulking status, histological subtype etc.

12. In the discussion (lines 437 – 442), the authors discuss the nature of phase I trial patients. It is important to note phase I trial populations are unrepresentative of the overall tumour population: patients who are well enough to tolerate up to 12 lines of prior treatment and still be alive up to 23 years after diagnosis are very atypical in ovarian cancer.

Minor points

1. Supplementary Figure 1A should possibly be moved to main figures so that that readers have a clear idea of treatment schedule without having to refer to the supplementary material.

2. Line 179 'OS and PFS data were retrieved from the electrical clinical trial records' – I think that this should be 'electronic' records.

3. Line 276 – ORR was... 20% (1/5) rather than '1/15' I think.

4. Patient 301-05 had had only 1 prior line of treatment (carboplatin and paclitaxel with no maintenance bevacizumab or PARPi) and was enrolled 15 months after diagnosis. Did this patient really have platinum-resistant disease? And it is most unusual to have 4 surgical interventions with only 1 line of chemotherapy.

5. Lines 325-329 states: 'Activity of TILT-123 in the ICI resistant setting was observed in patient 301-04 (cohort 2) with platinum resistant serious ovarian cancer, who previously received palliative pembrolizumab one year prior to enrolment. The patient showed a promising 36% reduction in injected tumor diameter at day 36, however the patient exhibited disease progression at the next visit and died 93 days after trial enrolment.' I'm afraid that death on day 93 with progressive disease between days 36 and next visit does not demonstrate 'activity of TILT-123'.

6. Lines 385-6: 'It is plausible that the resulting proteasome to immunoproteasome switch induced by TNF α , leads to increased stability of MHC class I and expression of endogenous peptides.' I think that this conjecture and should be removed.

Reviewer #3

(Remarks to the Author)

My review comments:

This is a single-arm, multicenter phase I dose escalation clinical trial that evaluated the safety and efficacy of combination therapy of oncolytic adenovirus TILT-123, an encoding tumor necrosis factor alpha (TNF α) and interleukin-2 (IL-2) designed to complement T-cell therapies, at varying doses with a immune checkpoint inhibition systemic therapy of pembrolizumab at a fixed dose of 200mg for patients with extensively pre-treated platinum-resistant or refractory ovarian cancer.

The key results:

The study enrolled 15 eligible patients to 4 escalating dosing cohorts. The treatment combination appears to be safe with the most common treatment related adverse events (AEs) reported as fatigue (46%), nausea (40%) and a decrease in lymphocyte count (40%), where 3 patients reported to have experienced grade 4 decreases in lymphocyte count.

Preliminary efficacy data showed overall median PFS of 3 months and OS of 6 months.

The overall disease control was reported to be 64% (9/14 evaluable patients) with best response being partial response. There was no indication of treatment dose response.

Notably, two patients who had failed multiple lines of prior therapy and presented with a large tumor burden showed survival benefits from the combination therapy, suggesting further investigation maybe warranted for such a combination therapy.

The study also performed preliminary yet extensive translational analyses, including pharmacokinetics, immunomodulation of tumors, as well as neutralizing antibodies (nAb), which may suggest a potential biomarker for treatment with TILT-123 and pembrolizumab.

Validity:

The study collected and presented the detailed clinical, safety and translational data from the enrolled patients. However, there are no aggregated patient characteristic and AE summaries overall and according to dosing cohorts. Both Table 1 and 2 are listings of individual characteristics and AE event counts respectively. There is also lack of clarity on how the dosing escalations were implemented (see below*). Figure 1 b and c included 'statistical' comparisons between the two groups (disease control or not), which is inappropriate as the disease control group status is correlated with the outcome measures of PFS and OS.

*Implementation of study design: the study is stated as a single-arm, phase I, 3+3 dose escalation design, with 4 cohorts enrolled patients, but it's unclear how the dose escalation was implemented; what exact (DLT?) criteria was used to guide the dosing escalation. What final dose(s) were recommended for the extension period to assess the treatment efficacy?

Significance:

The paper investigates a treatment strategy for heavily pre-treated as well as platinum refractory patients with ovarian cancer, who have very limited treatment options and unmet medical needs. The rationale of the combination therapy proposed were supported by the preclinical model data, as well as preliminary data from other studies that led to the hypothesis that TILT-123 may improve PD-1 efficacy by immunologically inducing immunosuppressive and / or innate tumors while PD-1 improve TILT-123 efficacy by delaying T-cell exhaustion yielding an additive effect. The results indicate the approach is reasonably safe and there are signals of possible efficacy that may be worth further investigation.

Data and methodology

No concerns about the study data are apparent. Some issues about the design and conduct of the study and presentation of results could use clarification (see below).

Analytical approach:

The analysis was mainly descriptive for the adverse events (AEs) assessment of treatment combination of TILT-123 + pembrolizumab.

For the exploratory efficacy assessment, the sample size (15 patients) is limited. Formal statistical inference such as log-rank test for the comparison of PFS and overall survival OS between groups of disease controls were inappropriate, as the outcomes and the comparison groups are intrinsically correlated. In addition, there was no prior power assessment for such statistical inference.

Suggested improvements:

DLT should be defined and specified in the paper – it is unclear what are the 'certain criteria' (line 154).

"DLT were defined as toxicities related to the study treatment that would prevent further administration of the agent at that dose level and would meet certain criteria."

Outcomes definitions:

Evaluation endpoint: unclear what this is if it seems to refer AE assessment timepoint at day 36(?)

Primary endpoint (definition?): it's noted as safety based on AEs measured on day 92. Unclear what AE grades were used to define this AE endpoint, per Adverse Events (AE) by (CTCAE) version 5.0.

There are a couple of secondary outcomes noted but without clear definition given:

e.g., what constitutes as a maximum tolerated dose (MTD), and how are PFS events defined? TTP is also mentioned in the results section but without an adequate definition.

Table 2 provided AE type by grades, overall counts and counts per patient. There is no breakdown by cohorts or TILT-123 doses. A tabular summary of counts of AEs by type and grade for each cohort should be included. Also, this type of summary would often include the maximum grade of each type for each patient, rather than all events.

Clarity and context:

A number of points about the methods and results need to be clarified, as discussed above. The explanation of context

seems reasonable, but that is outside my expertise.

References:

The study is based on a 3+3 dose escalation design, but the design criteria and the implementation were not provided. Given there are variations of 3+3 dose escalation strategy, the authors should provide reference(s) that help readers who are not familiar with the specific design features.

Reviewer expertise:

This review is mainly focused on the phase I study design and the corresponding AE and efficacy results presentation (biostatistician perspectives). The translational aspects of the study results, including pharmacokinetics, tumors immunomodulation, as well as assessment of neutralizing antibodies (nAb) as a potential biomarker are beyond the scope of my expertise and are not evaluated.

Minor comments:

The protocol (title) states this is a phase I/Ib study, however this paper only reports the phase I part of the study, this should be clearly stated as such.

Reviewer #4

(Remarks to the Author)

Version 1:

Reviewer comments:

Reviewer #1

(Remarks to the Author)

The authors have adequately addressed the reviewers' comments. I do not have any additional concerns.

Reviewer #2

(Remarks to the Author)

The authors have modified the manuscript in response to the initial reviewer comments.

1. The fact that some patients received intra-peritoneal TILT-123 is a fairly major change that really should have been mentioned clearly in the first version of the manuscript. Why did these patients receive intra-peritoneal rather than intratumoural injections? Lines 234-238 detail the treatment regimes but there is still no mention of the doses given by the IP route. Was IP dosing performed via an indwelling catheter?
2. Efficacy. I still do not believe that lesion-by-lesion response rates are an appropriate metric and thus I do not believe that the data presented in lines 282-287 are appropriate for a formal clinical study. Response or progression is by patient – thus the efficacy outcome is ORR of 7.1%, rising to 20% at the highest dose level. A simple statement that 'responses were observed in both injected and non-injected lesions' is possibly appropriate.
3. Descriptors such as 'notably' (lines 306, 321), notable (line 308), 'promising' (line 318) and 'Remarkably' (line 321) should be removed: the results are a factual presentation of the data.
4. Lines 353-356 state "Interestingly there was a drop in IL-2, hexon and fiber knob expression in the injected lesions from day 36 to 78, whilst expression increased in the non-injected lesions at the same time points. This dynamic likely indicates systemic dissemination of TILT-123 from injected lesions to non-injected tumors." However, in Figure 3b, there appears to be no measurement of any of these parameters in non-injected lesions after day 36. Thus, I do not believe that this statement is supported by the data presented.

Minor points

1. Y-axis labels in Figure 4a – "% CD8+, CD45+, PD1+, Epi+" – I think this should be Epi-?? Also what is the denominator in these graphs? All CD45+ cells? All CD8+ cells??
2. Was CD3 staining also performed to characterise NK cells? CD45+, CD56+ populations will include more than just NK cells (NKT cells, monocytes, dendritic cells etc) and a negative CD3 stain is also required to define NK cells.

Reviewer #3

(Remarks to the Author)

Overall reviewer comments:

The study team is applauded to provide additional information and clarifications for some of the comments and suggestions, including supplemental tab/figs for the proper study design, DLT definitions and agreement to drop the efficacy analysis/statistical inference.

However, there are remaining issues to be addressed (see below).

Primary (AE) endpoint definition:

AUTHORS RESPONSE

The following text has now been included in Design of the Methods section on page 4 line 177. (minor: could NOT find such info in the line #s to correspond to the info, but I found them elsewhere)

'AE grades were used to define this AE endpoint, per Adverse Events by NCI-CTCAE version 5.0 and if not possible to capture by this criteria the investigator will use AE criteria described in supplementary figure 4b.'

Reviewer response: the primary endpoint remains unclear: what (grades/types) counts as the primary AE endpoint among all the observed AEs, in addition to provide info on the grading method (thanks!).

AE summaries:

Reviewer responses: The AEs summary (primary endpoint) remain unclear and insufficient. It does not meet the standard of an oncology phase I trial AE reporting and need to be clarified.

Two tables are included to summarize the study AE profile: tab 2 and supplemental 3 (newly added), however,

- Tab 2 is a list of the treatment related AEs types by grades (what grades-maximum?) counts without knowing which cohorts were the events belong to.

- The Supl table 3 is very confusing. It's unclear what the #s mean (unique pts experience AEs?), nor is it consistent with the stated title/legend: there is no maximum grade info (?) and the #s (particularly for cohort 1) doesn't seem to make sense: why would cohort 1 have #s >=3 in the cells (if cohort 1 has only 3 pts, based on table 1)?

- As I noted it before: A tabular summary of counts of AEs by type and grade for each cohort should be included.

Evaluation endpoint: unclear what this is if it seems to refer AE assessment timepoint at day 36(?)

AUTHORS RESPONSE

This refers to the two imaging time points performed to evaluate treatment response. We have replaced 'evaluation endpoint' with 'first imaging time point'

Reviewer response: in manuscript Line 279: it notes 4/15 patients entered the extension period(cohort?): should it be 4/14 (from evaluation cohort or how was the analysis cohort defined?) Also, in figure 1a, it's still mention of 'evaluation endpoint' (?)

Statistical analysis:

Prior comment: Formal statistical inference such as log-rank test for the comparison of PFS and overall survival OS between groups of disease controls were inappropriate, as the outcomes and the comparison groups are intrinsically correlated. In addition, there was no prior power assessment for such statistical inference.

AUTHORS RESPONSE

As suggested, we have removed the statistical comparisons between the two groups in Figure 1b and c. We have removed the curves for disease control and no disease control to improve clarity and interpretability of Figure 1 for readers.

Reviewer response: The section didn't seem to be updated. Is log-rank test still relevant? if so where is being used and why?

What final dose(s) were recommended for the extension period to assess the treatment efficacy?

AUTHORS RESPONSE

'The phase 1b uses the virus and pembrolizumab dose from cohort 4. The final dose recommended for the next phase of trials evaluating efficacy has yet to be determined as the ongoing phase 1b is not completed yet'

This text has been added to the last lines of the Discussion on page 13 line 581.

Reviewer response: however, in fig 1a, there is a box to note that 4 pts are treated in extension cohort, who are they? from cohort 4? Then what (dose combinations?) were they receiving?

Again, the analysis cohort(s) for safety and efficacy endpoints should but was not clearly stated.

Results presentation:

1. Patient characteristics: There is no mention of who were in the extension cohort (which is noted n=4 from fig 1a). Are they a part of those pts enrolled in the (4) cohorts or a separate group?
2. Tab 1 and supl tab 1 have some overlap/redundant info (e.g, Histological subtype, Best overall response per RECIST1.1 or per IRECIST respectively), are they necessary? One summary tab that could be helpful (but missing) is the aggregated pt (demographic/disease) characteristics according to the dose cohorts
3. As I noted above, the AEs (primary endpoint) summary is unclear and inadequate and should be clarified.
4. Efficacy results (line 289-292): Is this relevant or should it be updated? Who are included in this analysis (cohort)? What/why tests/p-values, what are the proper interpretations? e.g, Line 289: what does it mean: 'Median OS was longer in the higher dose cohorts but not significantly'?"

A few typos/errors: I likely missed some. Please cross check the content with tabs/figs.

1. Figure 1a: Pg 6 line 238 states 6/14 met the primary endpoint at day 92, while figure 1a states 7/14 met the primary endpoint. What's the analysis cohort for the efficacy endpoint?
2. Line 268/pg7: Supplemental figure table 4 should be 'supplemental table 4'

Reviewer #4

(Remarks to the Author)

Version 2:

Reviewer comments:

Reviewer #2

(Remarks to the Author)

The authors have addressed my previous comments.

One small suggestion: on lines 92-93, the inclusion criteria state 'At least one tumor (>14 mm in diameter) or carcinomatosis must be available for local virus injection (i.t. and/or i.p.)'. This is ambiguous. I would suggest that they state that 'at least one tumor >14mm was required for i.t. injection. In patients with carcinomatosis not amenable to i.t. injection, i.p. injection was used', or something similar.

Reviewer #3

(Remarks to the Author)

Thanks for the thorough revisions and addressing my prior comments/questions. I have no further questions.

Response to referees

Reviewer #1 (Remarks to the Author): with expertise in ovarian cancer, (immuno) therapy, oncolytic viruses

Summary

This innovative early phase trial investigates a new, combinatorial approach to make immunologically 'cold' tumors 'hot'.

The authors' method involves an oncolytic adenovirus, previously investigated in early phase clinical trials (ref 1 - Pakola SA, Peltola KJ, Clubb JHA, Jirovec E, Hayboud L, Kudling TV, et al. Safety, efficacy, and biological data of T cell-enabling oncolytic adenovirus TILT-123 in advanced solid cancers from the TUNIMO monotherapy phase I trial. Clin Cancer Res. 2024), and anti-PD-1 checkpoint inhibitor pembrolizumab, which is in widespread clinical use. This is the first time these treatments have been used in combination.

There are several noteworthy findings from this study. The experimental intervention appeared safe and tolerable, with most significant toxicity limited to lymphopenia. Survival outcomes were not inferior (but also not superior) to those expected of pretreated, platinum resistance/refractory HGSC; particularly welcome was the finding of a partial response in a patient with mucinous carcinoma of the ovary for which there are limited therapeutic options. The combination approach appears to have elicited responses in both injected and non-injected tumors. There was an early finding of a putative biomarker in serum anti-adenovirus neutralizing antibodies (nAbs), an encouraging result with future studies in mind. This study, therefore, is original and would be of great interest to those working in gynecologic malignancies and cell therapies.

COMMENTS:

1) We note that the authors used CA-125 as a secondary endpoint and as a metric of progression when imaging was not available. To interpret the relevance of this as a metric, it would be useful to know whether all patients had elevated CA-125 at baseline and if the study adhered to established GCIG criteria for CA125 response. Only this standard GCIG criteria should be used to assess CA125 response. CA125 response is not a clinically meaningful endpoint in this recurrent disease setting. "CA125 levels decreased below baseline levels in 47 60% of the patients at least at one time point" - this should be removed as not a validated endpoint

AUTHORS RESPONSE

There was only one case where CA125 wasn't elevated at baseline and it was patient 301-11 - her values were within normal. As suggested, we have removed all CA125 data from the manuscript: Figure 1d, CA125 data in table 1, supplementary figure 1c and corresponding texts (mainly lines 278- 281 in results section and lines 38 and 46 of abstract) have been removed.

2) The exploratory objective, "evaluation of changes in patients' tumors or metastases and assessment of biological effects in biopsy samples" would be strengthened by some detail regarding what changes or biological effects are being referenced here.

AUTHORS RESPONSE

To strengthen the description of exploratory objectives we have included an additional follow-up sentence on page 5 line 205. *'Biological effects in biopsy samples included analysis of changes in proportions of different immune cells markers, expression of immune-related genes, as well as treatment related pathological changes.'*

3) *Intraperitoneal dosing of the experimental agent is mentioned in the introduction and methods but then is not referred to again in results or discussion, with mentions limited to intratumoral injection. Please clarify if any patients receive intraperitoneal injection of the agent?*

AUTHORS RESPONSE

We acknowledge the need to clarify the use of intraperitoneal dosing of TILT-123 in the report. Following an initial intravenous injection, the protocol allowed intratumoral or intraperitoneal (or both) injections on a patient by patient basis. We have therefore included additional text to highlight the three patients that received intraperitoneal injections. First, in *Safety of the Results* section on page 8 line 318, we have included the following sentence.

'Patients 302-05, 301-01 and 302-12 received three, four and two i.p injections (50ml total volume per dose) respectively. No serious adverse events related to treatment were reported amongst these patients. One case of grade 1 hyperkalemia was seen in patient 301-01 on day 36'

Secondly, in *discussion on page 12 line 524* we have included the following sentence.

'During the trial, three patients received i.p. dosing of TILT-123, all of which were well-tolerated. Patient 301-01 had grade 1 hyperkalemia on day 36, but it is unlikely that the i.p. injection was the sole contributing factor to this adverse event. Notably, at the same time point, the patient exhibited disease control, suggesting good tolerability and supporting the feasibility of i.p. injection. Overall, i.t. and i.p. injections were well-tolerated by patients and although requiring some skill, local delivery of oncolytic virus was performed successfully without long-term implications to patient well-being'

4) *Did any patients suffer adverse events as a direct result of injection into the tumor (for example, the patient with hemoperitoneum grade 3) as distinct from systemic effects of the treatment? It may be of benefit to clarify this and also to report if there were any challenges in local delivery of the oncolytic virus.*

AUTHORS RESPONSE

To report the experience of using i.t. injections and local delivery of oncolytic virotherapy, we have included the following text in *Safety of the Results* section on page 8 line 322.

*'Analysis of adverse events related to i.t. injection included two grade 1 infusion site reactions related to TILT-123 in patients 301-03 and 301-04 (table 2). In addition, patients 301-03 and 302-07 reported two incidences of grade 2 abdominal pain on day 8 caused by activity of TILT-123. Five incidences of pain following injection site/biopsy are reported in **Supplementary figure Table 4** and were not related to treatment. Grade 3 hemoperitoneum was reported in patient 301-12, as related to injection of TILT-123, and not the activity of TILT-123. The patient required prolonged hospitalization but was eventually discharged in stable condition.'*

In addition we have included the following text to the *Discussion* on page 12 line 527.

'Overall, i.t. and i.p. injections were well-tolerated by patients and although requiring some skill, local delivery of oncolytic virus was performed successfully without long-term implications to patient well-being.'

5) *How was platinum resistance and refractory defined?*

AUTHORS RESPONSE

Platinum refractory was defined as progression of cancer within 30 days of the most recent dose of cisplatin or carboplatin (in the absence of prior response or disease control). Platinum resistant was defined as progression of cancer within 183 days of the most recent dose of cisplatin or carboplatin. These definitions have been included in *Patients and Methods* of the *Methods* section on page 3 line 148.

6) *Throughout the results section would try to avoid including conclusions and report only the results.*

AUTHORS RESPONSE

The following conclusions have been removed from the results section and either incorporated into the discussion or omitted from the manuscript.

- a finding consistent with a higher dose causing more pseudo-progression, commonly reported with oncolytic viruses (Supplementary Figure 1f) (25). For example, initial progression as seen in 54% of melanoma patients later responding to an oncolytic herpes virus (26). Pseudoprogession is caused by virus association inflammation and accumulation of leukocytes. This phenomenon has also been reported for immune checkpoint inhibitors, but is much stronger with highly immunogenic agents such as oncolytic viruses (27, 28).
- Mallory bodies (here seen in tumors) are traditionally associated with liver diseases such as cirrhosis, and mechanistically caused by proteasome stress induced by TNF α (expressed by TILT-123), resulting in an accumulation of misfolded and oxidized proteins (29). It is plausible that the resulting proteasome to immunoproteasome switch induced by TNF α , leads to increased stability of MHC class I and expression of endogenous peptides (30).
- This increase in immune activity coincided with an increase in tumor diameter in CT which further supports the notion of inflammatory pseudoprogession (Figure 1e – 1h).
- Altogether these data demonstrate TILT-123 can systemically disseminate and modulate the tumor immune microenvironment of ovarian cancer, and reveals early signs of a potential biomarker related to the activation of antibody producing cells.

7) *The median OS is 27 weeks and this would be within the expected survival range for this patient population. Would therefore consider omitting this "Notably, more than half of the patients survived longer than 6 months, which is encouraging in a highly pre-treated population with advanced platinum resistant or refractory ovarian cancer." as the findings are not notable and if included I would report 95% confidence intervals to clarify*

AUTHORS RESPONSE

We have decided to omit this sentence from the manuscript as suggested.

8) Line 62 *"The quantification of neutralizing antibodies against adenovirus provides a preliminary measure to predict the efficacy of treatment. Investigation of TILT-123, pembrolizumab"*

It is not clear how clinicians would apply this in clinical practice and why RECIST response would not instead be used to assess response. What is the turn around time for this assay and would it be readily available?

AUTHORS RESPONSE

We would not propose to use nabs measurement instead of RECIST1.1 but as an independent biomarker for efficacy and especially survival. The current data (Figure 4c-f) demonstrates that the presence of neutralizing antibodies against adenovirus at baseline predicts longer overall survival but not RECIST1.1 response, whilst a higher titer post-treatment predicted both longer overall survival and RECIST1.1 response. The caveat of RECIST1.1 in oncolytic immunotherapy is the pseudoprogression regularly seen (1, 2, 3). Therefore, additional biomarkers of survival would be welcome. Looking forward, we intend to study these issues further with a larger patient number in the next phase of trials.

In terms of clinical implementation, the assay turnaround time is three days, or one week if taking into consideration cell culture and it requires a blood (serum) sample. We have added text in *Serum Anti-adenovirus antibodies in Methods* on page 6 line 230.

- *'In brief, A549 were seeded overnight in 96-well plates and the next day serial dilutions of complement inactivated serum samples from patients were incubated with Ad5/3-luciferase for 1 hour. A549 cells were then infected with Ad5/3-luciferase pre-incubated with serum overnight and luciferase expression quantified the following day using Luciferase Assay System (Promega, WI, USA). The assay can be easily set up in any laboratory and takes 3 days assuming cells are kept in culture'*

9)Line 276 *'The overall response rate (ORR) was 7.1% (1/14) and 20% (1/15) at the highest dose level.'* Is there a typo should it be 1/5 to equate to 20%.

AUTHORS RESPONSE

We have corrected the typo to 1/5 and not 1/15.

10) *The methods describe lesion disease control rate but in the results "Disease control rate" is reported so it needs to be clarified if they are referring to RECIST "SD and CR/PR for targets and non-targets" or are they referring to the response in the injected lesions. This should be more clearly defined in the methods and the results- was this a pre-specified secondary endpoint?*

AUTHORS RESPONSE

In the current text, we provided disease control rate and overall/objective response rate as per patient and per lesion using the same criteria (DCR = CR + PR + SD, ORR = CR + PR). L-DCR and L-ORR are defined in *assessment of treatment response* in the *methods* section and values are presented within Figures 1e-1h for both injected and

non-injected. However, we have now included the following text in assessment of treatment response of Methods section on page 6 line 220.

'For both patient and lesion response analysis the following criteria were used to define DCR and ORR. Evaluation of lesion specific disease control rate was not a pre-specified secondary endpoint.'

DCR = CR + PR + SD,

ORR = CR + PR

11) Line 311 - *"The patient came off the trial but surprisingly survived for 434 days from enrolment. Such cases are compatible with pseudoprogression resulting in treatment interruption despite anti-tumour benefits" Did the patient receive any subsequent therapy? If so then this sentence must be reworded as it is misleading and any OS benefit could be attributed to a different anticancer intervention*

AUTHORS RESPONSE

The patient did receive treatment after coming off the trial and we agree this sentence is potentially misleading. We have therefore modified the sentence as follows.

'The patient came off the trial on day 92 and started gemcitabine and topotecan to treat metastatic disease on days 128 and 247 respectively. Altogether, the patient survived 434 days after enrolment.'

Reviewer #2 (Remarks to the Author): with expertise in ovarian cancer, therapy, oncolytic viruses

In this study, the authors present data from a phase I trial of an oncolytic adenovirus Ad5/3 E2F-D24-hTNF α -IRES-hIL-2 with pembrolizumab in patients with recurrent ovarian cancer. The vector is a chimeric Ad5/3 backbone with a 24bp deletion in the E1A CR2 region and E1 expression driven by an E2F promoter. The vector also encodes two transgenes, TNF α and IL-2. The first injection was given IV, with subsequent injections given IT. Pembrolizumab was added from day 36 to allow assessment of virus-only activity and safety. There were 4 dose cohorts of increasing viral dose; pembrolizumab dose was constant. There were only two formal assessment points, days 36 and 92.

The headline data are that treatment was safe, with no DLT. The TEAE are as one would expect from prior trials of oncolytic adenovirus and also pembrolizumab with no suggestion of increased pembrolizumab toxicity. There was no cytokine release syndrome but an interesting drop in lymphocyte counts after the first virus dose, and subsequent oscillations in both lymphocyte and neutrophil counts.

There was one partial response in a patient with mucinous carcinoma. In all other patients, best response (by RECIST) was SD or PD. Three patients, including the one who had had a partial response, were alive at data cutoff.

There was extensive translational research, with multiple biopsies and blood samples, for which the authors (and more importantly the participants) should be congratulated. Key points were possible detection of Hexon and Fiber knob mRNA in uninjected lesions, suggesting infection of those lesions. There were also some suggestions of immune cell infiltration following injections, although the true meaning of these changes is not clear.

Overall, this type of trial is highly challenging, and the authors have undertaken a huge amount of work to complete the study. However, the participant number is low at 15, making it hard to reach definitive conclusions.

Major points:

1. *Patients and methods - Table 1 and patient characteristics. Critical features for these patients, including BRCA1/2 mutation status, MMR status, are required. For the patients labelled as 'serous' (301-01, 301-03, 301-04), it is important to identify whether they had high or low grade serous disease –simple IHC should be able to answer that question. This is of relevance because LGSOC often grows very slowly with very long periods of stable disease and long overall survival (patient 302-06 being a very clear example).*

AUTHORS RESPONSE

We have modified table 1 to define low grade or high-grade serous status in patient 301-01, 301-03 and 301-04. All three patients presented with HGSOE.

We have also included BRCA1/2 mutation, HDR and MSI status for those patients which information are available. These data are not always collected for ovarian cancer patients at our sites.

2. *Assessment of treatment response. The authors use unusual definitions of DCR and ORR – 'lesion' control rate and objective response rate. Are the authors excluding non-injected or non-target lesions here? The wording in the results text should ORR by RECIST1.1 and by iRECIST using the standard definitions (as shown in Table 1). Strictly speaking, DCR is not a RECIST term, so the authors should state clearly how they defined this (I think they used all patients who achieved PR and SD as best RECIST1.1 response)*

AUTHORS RESPONSE

In the current text, we provided disease control rate and overall/objective response rate as per patient and per lesion using the same criteria (DCR = CR + PR + SD, ORR = CR + PR). L-DCR and L-ORR are defined in *assessment of treatment response* in the *methods* section and values are presented within Figures 1e-1h for both injected and non-injected. However, we have now included the following information on page 5 line 220.

'For both patient and lesion response analysis the following criteria were used to define DCR and ORR. Evaluation of lesion specific disease control rate was not a pre-specified secondary endpoint.

DCR = CR + PR + SD

ORR = CR + PR'

CT analysis reported all but two non-target lesions (one in extension treatment period) as "present without unequivocal progression" as they were too small to measure. For this reason, we cannot define these using RECIST1.1 terms, and RECIST1.1 evaluability was not an inclusion criterion in this safety study. We reported these data in supplementary figure 1c and 1d, but have now decided to remove these sections from the manuscript to improve clarity for the readers and overall message.

3. *Figure 1b and c should be presented as single lines – ie PFS and OS without the 'disease control' and 'no disease control' lines. This is particularly true for PFS – by definition, patients*

who have no disease control on day 92 will have a poorer PFS than those who have achieved disease control on day 92. A spider plot of individual patient data over time would also be helpful.

AUTHORS RESPONSE

As suggested, we have presented figure 1b and c as single lines without 'disease control' and 'no disease control'.

We have also included a spider plot for injected and non-injected tumours in Figure 1e and updated the figure legends accordingly

4. Similarly, Figures 1e-h should be presented as a single waterfall plot showing overall change compared to baseline for the whole cohort. It might be possible to present data from day 36 and day 92 separately in the supplementary data.

AUTHORS RESPONSE

As suggested, we have re-formatted figure 1e-h as a single waterfall plot showing overall change from baseline and moved separate time points to supplementary figure 1b. The figure legends have been updated accordingly.

5. The efficacy section states that a 'decrease in CA125 levels was observed in 60% of patients at least at one time point'. This is not a meaningful measure. What was the CA125 response rate by GCIG criteria? From the data presented in Figure 1d, I think that the CA125 response rate was 0%. This may not matter given the inflammatory nature of adenovirus infection, but the authors should use standard measurements.

AUTHORS RESPONSE

As suggested, we have removed CA125 data from the manuscript. Figure 1d, CA125 data in table 1, supplementary figure 1c and corresponding texts (mainly lines 342 in results section and lines 48 of abstract) have been omitted.'

6. Similarly, on lines 286-287, the authors state 'Evaluation of non-target lesions revealed only 1 out of 18 (6%) (Patient 301-10) progressing on day 36 and 92'. RECIST evaluation is a sum of all lesions (target and non-target), so discussing non-target lesions is not terribly helpful or meaningful.

AUTHORS RESPONSE

Agreed, that section on page 8 and 9 line 351 and supplementary figure 1c and d have been deleted.

7. The authors also discuss 'pseudoprogession'. This term requires lesions to get bigger since baseline but then to reduce in size. If a lesion reaches the RECIST definition of PD and then stays at the same size on the subsequent scan, that is iUPD rather than pseudoprogession. However, given that there was only 1 partial response and only two assessment points after the start of treatment, I am not sure how Supplementary Figure 1f can be plotted to show Dose Pseudoprogession.

AUTHORS RESPONSE

As suggested we have omitted supplementary figure 1F and corresponding discussion on page 9 line 360 from the manuscript.

8. *Figure 3b. The possible (ie non-significant) demonstration of hexon and fiber in non-injections lesions is of great interest. However, what is the lower limit of detection here? In Figure 3a, the data are presented as VP/ml with LLOQ between 10¹ and 10². However, in 3b, the y-axis units are 'log₂(Counts)'. It would be hard to imagine that there would be any detectable Hexon or Fiber knob mRNA or indeed IL2 and TNF using vector-specific primers on day -14 in any lesion, injected or uninjected.*

AUTHORS RESPONSE

The data presented in Figure 3a and 3b were calculated using two different assays. The latter of which was analysed using nanostring ncounter assay where LLOQ are not provided by the service provider. Vector-specific primers for IL2 were only used for the qPCR assay in Figure 3a, whereas IL2 and TNF were detected using standard commercially available primers in the nanostring assay. We agree that it is not expected to detect hexon and fiber knob mRNA on day-14 in either injected or uninjected lesions. To improve clarity of the figure, we have changed the graphs to show mRNA abundance normalized to baseline (day -14) and updated the description in *Gene expression profiling* in *Methods* section on page 6 line 267. The figure legends have been updated accordingly.

9. *Figure 3d – the white dotted arrows are supposed to show cytopathic effect, which is very far from obvious. Similarly, the black arrows are supposed to present viral inclusion bodies, which are again far from clear. Several of the black arrows point to interstitial space. There is also discussion of Mallory bodies, which I believe are hallmarks of liver disease, especially of alcoholic origin. Was a formal pathology review performed? And was the pathologist blinded to whether the sample was injected, uninjected, pre- or post-treatment?*

AUTHORS RESPONSE

We have improved the clarity of Figure 3d and adjusted the corresponding text according to further advice provided by an experienced pathologist. The pathologist identified basophilic inclusions, microcalcifications, psammoma bodies and scarce inflammation in sample shown in Figure 3d. The pathologist was not blinded to whether the sample was pre or post treatment but she was blinded to whether the sample was injected or noninjected. The pathologist has been added as an author due to her contribution to this manuscript. In addition, we have added the following text to *IHC and multiplex immunofluorescence staining* in *Methods* on page 6 line 260.

'H&E images were evaluated for possible treatment-induced changes by an experienced pathologist'

10. *Figure 3e is reported to show co-localisation of virus positive regions with areas of high immune cell infiltration, represented by an increase in CD8+, CD4+ and 372 CD20+, as well as a decrease in PD-L1+ immune clusters. There is no demonstration of virus positivity in Figure 3e and there is no formal analysis to correlate CD8 etc cell density with Hexon staining. However, I do note the finding of an increase in CD8 and NK cells in Figure 4a.*

AUTHORS RESPONSE

The regions presented in Figure 3e are matched to the regions presented in Figure 3c, which show anti-adenovirus hexon IHC. This has been described in the figure legend.

The aim of this figure is to present descriptive analysis of one interesting case while overall effects observed in the tumours can be found in Figure 4a.

11. *The humoral response data are very interesting, in particular the finding that those with PD had lower maximum post-treatment nAb titers, suggesting that those patients who cannot mount an Ab response to large virus doses have a poorer outcome. In terms of the KM curves in Figures 4a and f, these need to be regarded cautiously given the multiple factors that can determine overall survival, including age, prior treatments, debulking status, histological subtype etc.*

AUTHORS RESPONSE

We agree that multiple factors can influence overall survival including those mentioned by the Reviewer. Indeed fitness of humoral immunity may represent other factors such as overall health or age. We have added the following text in the *discussion* on page 13 line 555 detailing this point: *'Additionally, interpretation of the nAb data should be regarded with caution as multiple factors can influence overall survival, including overall health and age which play a role in the status of humoral immunity'*.

12. *In the discussion (lines 437 – 442), the authors discuss the nature of phase I trial patients. It is important to note phase I trial populations are unrepresentative of the overall tumour population: patients who are well enough to tolerate up to 12 lines of prior treatment and still be alive up to 23 years after diagnosis are very atypical in ovarian cancer.*

AUTHORS RESPONSE

Agreed. We have added the following text at the end of lines 521 to highlight this point. *'Additionally, the patients reported in this study may not be representative of the average patient population, for example patient 302-06 who was still alive 23 years after initial diagnosis..'*

Minor points

1. *Supplementary Figure 1A should possibly be moved to main figures so that that readers have a clear idea of treatment schedule without having to refer to the supplementary material.*

AUTHORS RESPONSE

As suggested, we have moved supplementary Figure 1A to the main figure to improve clarity for the readers.

2. *Line 179 'OS and PFS data were retrieved from the electrical clinical trial records' – I think that this should be 'electronic' records.*

AUTHORS RESPONSE

We have corrected this to 'electronic' as suggested.

3. *Line 276 – ORR was... 20% (1/5) rather than '1/15' I think.*

AUTHORS RESPONSE

We have corrected the typo to 1/5 and not 1/15.

4. Patient 301-05 had had only 1 prior line of treatment (carboplatin and paclitaxel with no maintenance bevacizumab or PARPi) and was enrolled 15 months after diagnosis. Did this patient really have platinum-resistant disease? And it is most unusual to have 4 surgical interventions with only 1 line of chemotherapy.

AUTHORS RESPONSE

We checked with the investigator that indeed 301-05 had platinum resistant disease. Regarding surgical interventions, we checked with the investigators and found out that there was a typographical error and in fact patient 301-05 had one line of surgery (10 months prior to study entry: interval cytoreductive operation). This has now been updated in the corresponding text and tables.

5. Lines 325-329 states: 'Activity of TILT-123 in the ICI resistant setting was observed in patient 301-04 (cohort 2) with platinum resistant serious ovarian cancer, who previously received palliative pembrolizumab one year prior to enrolment. The patient showed a promising 36% reduction in injected tumor diameter at day 36, however the patient exhibited disease progression at the next visit and died 93 days after trial enrolment.' I'm afraid that death on day 93 with progressive disease between days 36 and next visit does not demonstrate 'activity of TILT-123'.

AUTHORS RESPONSE

We have removed 'Activity of' from line 394 as to not suggest treatment efficacy but rather report a potentially interesting patient case.

6. Lines 385-6: 'It is plausible that the resulting proteasome to immunoproteasome switch induced by TNF α , leads to increased stability of MHC class I and expression of endogenous peptides.' I think that this conjecture and should be removed.

AUTHORS RESPONSE

We have omitted this sentence from the manuscript.

Reviewer #3 (Remarks to the Author): with expertise in biostatistics, clinical trial study design

My review comments:

This is a single-arm, multicenter phase I dose escalation clinical trial that evaluated the safety and efficacy of combination therapy of oncolytic adenovirus TILT-123, an encoding tumor necrosis factor alpha (TNF α) and interleukin-2 (IL-2) designed to complement T-cell therapies, at varying doses with a immune checkpoint inhibition systemic therapy of pembrolizumab at a fixed dose of 200mg for patients with extensively pre-treated platinum-resistant or refractory ovarian cancer.

The key results:

The study enrolled 15 eligible patients to 4 escalating dosing cohorts. The treatment combination appears to be safe with the most common treatment related adverse events

(AEs) reported as fatigue (46%), nausea (40%) and a decrease in lymphocyte count (40%), where 3 patients reported to have experienced grade 4 decreases in lymphocyte count.

Preliminary efficacy data showed overall median PFS of 3 months and OS of 6 months. The overall disease control was reported to be 64% (9/14 evaluable patients) with best response being partial response. There was no indication of treatment dose response. Notably, two patients who had failed multiple lines of prior therapy and presented with a large tumor burden showed survival benefits from the combination therapy, suggesting further investigation maybe warranted for such a combination therapy.

The study also performed preliminary yet extensive translational analyses, including pharmacokinetics, immunomodulation of tumors, as well as neutralizing antibodies (nAb), which may suggest a potential biomarker for treatment with TILT-123 and pembrolizumab.

Validity:

The study collected and presented the detailed clinical, safety and translational data from the enrolled patients. However, there are no aggregated patient characteristic and AE summaries overall and according to dosing cohorts. Both Table 1 and 2 are listings of individual characteristics and AE event counts respectively.

AUTHORS RESPONSE

We have now summarised AE according to dosing cohorts in supplementary table 3.

There is also lack of clarity on how the dosing escalations were implemented (see below).*

AUTHORS RESPONSE

We have improved the description of how dose escalation was implemented in *Design of the Methods* on page 4 lines 169-179 and included a graphical description in supplementary figures (see below for more information).

Figure 1 b and c included 'statistical' comparisons between the two groups (disease control or not), which is inappropriate as the disease control group status is correlated with the outcome measures of PFS and OS.

AUTHORS RESPONSE

As suggested, we have removed the statistical comparisons between the two groups in Figure 1b and c. In fact, we have decided to remove the lines (disease control or not) in their entirety from Fig 1 to improve clarity.

**Implementation of study design: the study is stated as a single-arm, phase I, 3+3 dose escalation design, with 4 cohorts enrolled patients, but it's unclear how the dose escalation was implemented; what exact (DLT?) criteria was used to guide the dosing escalation.*

AUTHORS RESPONSE

As suggested, we have included the following text describing how dose escalation was implemented in *Design of Methods* on page 4 lines 169- 179 and included the following graphical description in supplementary figures. *'DLT is defined to be a toxicity event that prevents further administration of the agent at that dose level. In particular, DLT is specifically defined as any*

death not clearly due to the underlying disease or extraneous causes, or as any treatment-related Grade ≥ 3 non-hematologic AE, or any Grade ≥ 4 hematologic AE according to the NCI-CTCAE v 5.0, occurring during the DLT window of observation period. The DLT window of observation will be from day 1 to day 57, covering from the first TILT-123 injection up to 21 days after the first dosage of TILT-123 in combination with pembrolizumab'

The decision tree for the 3+3 dose escalation design and DLT implementation can now be found in supplementary figure 4a and 4b'

Supplementary figure 4a Decision tree for dose escalation

The occurrence of any of the following toxicities during the first cycle of treatment with TILT-123 and pembrolizumab will be considered a DLT, if assessed by the investigator to be related to any of the study treatments:

1. Grade 4 nonhematologic toxicity (not laboratory).
2. Grade 4 hematologic toxicity lasting ≥ 7 days, except thrombocytopenia:
 - a. Grade 4 thrombocytopenia of any duration
 - b. Grade 3 thrombocytopenia associated with clinically significant bleeding
3. Any nonhematologic AE \geq Grade 3 in severity should be considered a DLT, with the following exceptions: Grade 3 fatigue lasting ≤ 3 days; Grade 3 diarrhea, nausea, or vomiting without use of anti-emetics or anti-diarrheals per standard of care; Grade 3 rash without use of corticosteroids or anti-inflammatory agents per standard of care.
4. Any Grade 3 or Grade 4 non-hematologic laboratory value if:
 - a. Clinically significant medical intervention is required to treat the participant or
 - b. The abnormality leads to hospitalization, or
 - c. The abnormality persists for >1 week.
 - d. The abnormality results in a Drug induced Liver Injury (DILI)
 - e. Exceptions: Clinically non-significant, treatable, or reversible laboratory abnormalities including liver function tests, uric acid, etc.
5. Febrile neutropenia Grade 3 or Grade 4 lasting more than 7 days and with additional presence of clinical symptoms [59]:

- a. Grade 3 is defined as ANC <1000/mm³ with a single temperature of >38.3 degrees C (101 degrees F) or a sustained temperature of ≥38 degrees C (100.4 degrees F) for more than 1 hour
 - b. Grade 4 is defined as ANC <1000/mm³ with a single temperature of >38.3 degrees C (101 degrees F) or a sustained temperature of ≥38 degrees C (100.4 degrees F) for more than 1 hour, with life-threatening consequences and urgent intervention indicated.
6. Prolonged delay (>2 weeks) in initiating the second dosage of study-treatment due to treatment-related toxicity.
 7. Any treatment-related toxicity that causes the participant to discontinue treatment from day 1 to day 57 (or day 43, for the unacceptable toxicity definition).
 8. Missing >25% of TILT-123 doses as a result of drug-related AE(s) during the first cycle.
 9. Grade 5 toxicity.
 10. A study intervention-related treatment-emergent adverse event (TEAE) that in the opinion of the sponsor and or investigators is of potential clinical significance such that further dose escalation would expose participants to unacceptable risk.

Supplementary figure 4b List of toxicities used to define a DLT if present during the first cycle of treatment.

What final dose(s) were recommended for the extension period to assess the treatment efficacy?

AUTHORS RESPONSE

'The phase 1b uses the virus and pembrolizumab dose from cohort 4. The final dose recommended for the next phase of trials evaluating efficacy has yet to be determined as the ongoing phase 1b is not completed yet'

This text has been added to the last lines of the *Discussion* on page 13 line 581.

Significance:

The paper investigates a treatment strategy for heavily pre-treated as well as platinum refractory patients with ovarian cancer, who have very limited treatment options and unmet medical needs. The rationale of the combination therapy proposed were supported by the preclinical model data, as well as preliminary data from other studies that led to the hypothesis that TILT-123 may improve PD-1 efficacy by immunologically inducing immunosuppressive and / or innate tumors while PD-1 improve TILT-123 efficacy by delaying T-cell exhaustion yielding an additive effect. The results indicate the approach is reasonably safe and there are signals of possible efficacy that may be worth further investigation.

Data and methodology

No concerns about the study data are apparent. Some issues about the design and conduct of the study and presentation of results could use clarification (see below).

Analytical approach:

The analysis was mainly descriptive for the adverse events (AEs) assessment of treatment combination of TILT-123 + pembrolizumab.

AUTHORS RESPONSE

To improve clarity for the reader, we have included the following text in *Outcomes of Methods* section.

'Assessment of adverse events (AEs) for the combination of TILT-123 + pembrolizumab was descriptive.'

For the exploratory efficacy assessment, the sample size (15 patients) is limited.

AUTHORS RESPONSE

Agreed, the following statement has been added to the *Discussion* on page 12 line 518.

'With that said, the sample size in the present dose escalation study is too limited (total n= 15; n=5 imaged patients at the highest dose) to assess efficacy reliably. Therefore a larger cohort is required to be able to adequately evaluate response with sufficient statistical power.'

Formal statistical inference such as log-rank test for the comparison of PFS and overall survival OS between groups of disease controls were inappropriate, as the outcomes and the comparison groups are intrinsically correlated. In addition, there was no prior power assessment for such statistical inference.

AUTHORS RESPONSE

As suggested, we have removed the statistical comparisons between the two groups in Figure 1b and c. We have removed the curves for disease control and no disease control to improve clarity and interpretability of Figure 1 for readers.

Suggested improvements:

DLT should be defined and specified in the paper – it is unclear what are the 'certain criteria' (line 154).

"DLT were defined as toxicities related to the study treatment that would prevent further administration of the agent at that dose level and would meet certain criteria."

AUTHORS RESPONSE

'DLT is defined to be a toxicity event that prevents further administration of the agent at that dose level. In particular, DLT is specifically defined as any death not clearly due to the underlying disease or extraneous causes, or as any treatment-related Grade ≥ 3 non-hematologic AE, or any Grade ≥ 4 hematologic AE according to the NCI-CTCAE v 5.0, occurring during the DLT window of observation period. The DLT window of observation will be from day 1 to day 57, covering from the first TILT-123 injection up to 21 days after the first dosage of TILT-123 in combination with pembrolizumab' This information has been added to *Design of Methods* section on page 4 line 169. We hope that this, in addition to the newly added supplementary figure 4a-4d clarifies how DLT was defined and utilized in this trial.

Outcomes definitions:

Evaluation endpoint: unclear what this is if it seems to refer AE assessment timepoint at day 36(?)

AUTHORS RESPONSE

This refers to the two imaging time points performed to evaluate treatment response. We have replaced 'evaluation endpoint' with 'first imaging time point'

Primary endpoint (definition?): it's noted as safety based on AEs measured on day 92. Unclear what AE grades were used to define this AE endpoint, per Adverse Events (AE) by (CTCAE) version 5.0.

AUTHORS RESPONSE

The following text has now been included in *Design* of the *Methods* section on page 4 line 177.

'AE grades were used to define this AE endpoint, per Adverse Events by NCI-CTCAE version 5.0 and if not possible to capture by this criteria the investigator will use AE criteria described in supplementary figure 4b.'

The following text extracted from the protocol are now included as supplementary figure 4b.

The trial investigator will rate the severity of each AE according to the NCI CTCAE guideline version 5.0.

For AEs not possible to capture using the NCI CTCAE, the investigator will use the following definitions to rate the severity of each AE:

- Grade 1 – Mild; asymptomatic or mild symptoms; clinical or diagnostic observations only; intervention not indicated.
- Grade 2 – Moderate; minimal, local or non-invasive intervention indicated; limiting age-appropriate instrumental activities of daily living. Instrumental activities of daily living refer to preparing meals, shopping for groceries or clothes, using the telephone, managing money, etc.
- Grade 3 – Severe or medically significant but not immediately life-threatening; hospitalization or prolongation of hospitalization indicated; disabling; limiting self-care activities of daily living. Self-care activities of daily living refer to bathing, dressing and undressing, feeding self, using the toilet, taking medications, and not bedridden.
- Grade 4 – Life-threatening consequences; urgent intervention indicated. Note, for safety reporting purposes, grade 4 "Life-threatening" is not considered identical to the ICH seriousness criterion "Life-threatening". The investigator must state "life-threatening" as the seriousness criteria on the SAE form in order to be considered life-threatening.
- Grade 5 – Death related to AE.

The grade assigned by the investigator should be the most severe, which occurred during the AE period.

There are a couple of secondary outcomes noted but without clear definition given: e.g., what constitutes as a maximum tolerated dose (MTD), and how are PFS events defined? TTP is also mentioned in the results section but without an adequate definition.

AUTHORS RESPONSE

The following text extracted from the protocol have been included in supplementary figure 4c and is referred to in *Outcomes* of the *Methods* section on page 5 line 209.

- The dosing of this new MTD cohort will be according to the following rules: (1) if both DLT events were seen before day 8, the MTD is half a log (3.3) lower for the i.v. dose but the i.t./i.p. dose remains the same, (2) if both DLT events were seen before day 36 (but both were not seen before day 8), the MTD is half a log (3.3) lower for the both the i.t./i.p. and the i.v. doses, (3) in all other scenarios, the MTD is half a log (3.3) lower for the i.t./i.p. dose while the i.v. dose remains the same.

Of note, when dose-escalation reaches cohort 4 and a DLT occurs in 2/5 patients at that dose level, an MTD cohort will be according to the following rules: (1) if both DLTs are seen before day 8, the MTD is the cohort 3 i.v. dose and the i.t./i.p. dose remains the same, (2) if both DLTs are seen before day 36, the MTD is the cohort 3 i.t./i.p. and i.v. doses, (3) if both DLTs occur in all other scenarios, the MTD is the cohort 3 i.t./i.p. dose and the i.v. dose remains the same. Pembrolizumab dose will remain the same despite lowering the dose of TILT-123.

- PFS was defined by time from first dose of TILT-123 to the first documented disease progression by CT (RECIST1.1) or death due to any cause, whichever occurs first. If a patient had not had an event (progression or death) before end of trial, then PFS will be censored at the last date known of non-progression.
- TTP was defined by time from first dose of TILT-123 to the first documented disease progression by CT (RECIST1.1) or death. If progression or death was not related to disease then PFS will be censored.

Table 2 provided AE type by grades, overall counts and counts per patient. There is no breakdown by cohorts or TILT-123 doses. A tabular summary of counts of AEs by type and grade for each cohort should be included. Also, this type of summary would often include the maximum grade of each type for each patient, rather than all events.

AUTHORS RESPONSE

We have now summarised AE according to type and maximum grade (reported per patient) by dosing cohorts in supplementary table 3.

Clarity and context:

A number of points about the methods and results need to be clarified, as discussed above.

AUTHORS RESPONSE

Please see above for our responses, which include those addressing points regarding clarification and context of methods and results.

The explanation of context seems reasonable, but that is outside my expertise.

References:

The study is based on a 3+3 dose escalation design, but the design criteria and the implementation were not provided. Given there are variations of 3+3 dose escalation strategy, the authors should provide reference(s) that help readers who are not familiar with the specific design features.

AUTHORS RESPONSE

As suggested above, we have included text describing how dose escalation was implemented in *Design of Methods* on page 4 line 169 and included the graphical description (Decision tree for dose escalation) as supplementary figure 4a. We have also included reference to the original publications from which the design was described and developed (4, 5, 6)

Reviewer expertise:

This review is mainly focused on the phase I study design and the corresponding AE and efficacy results presentation (biostatistician perspectives). The translational aspects of the study results, including pharmacokinetics, tumors immunomodulation, as well as assessment of neutralizing antibodies (nAb) as a potential biomarker are beyond the scope of my expertise and are not evaluated.

Minor comments:

The protocol (title) states this is a phase I/Ib study, however this paper only reports the phase I part of the study, this should be clearly stated as such

AUTHORS RESPONSE

We have changed the following title and paragraphs of the manuscript to specify phase 1a.

- Title
 - Safety, efficacy and biological data from phase 1a of PROTA: A Phase I trial combining oncolytic adenovirus TILT-123 with Pembrolizumab for the treatment of patients with Platinum Resistant or Refractory Ovarian cancer
- Abstract (patients and methods)
 - Phase 1a of PROTA (NCT05271318) was a single-arm, multicenter dose escalation trial designed to
- Line 136
 - In this article we report the results from the phase 1a part of PROTA, a single-arm, multicenter phase I dose escalation clinical trial of combined i.v. and i.t./i.p. injection of TILT-123, combined with systemic pembrolizumab for patients with platinum resistant or refractory ovarian cancer.

Reviewer #4 (Remarks to the Author):

AUTHORS RESPONSE

We are grateful for your contributions to this peer review.

References

1. Conley, A., Larson, C., Oronsky, B., Stirn, M., Caroen, S. and Reid, T.R., 2024. Hypothesis: AdAPT-001 and pseudoprogression—when seeing is not necessarily believing. *Journal for ImmunoTherapy of Cancer*, 12(6).
2. Zamora, C., Lopez, M., Cunningham, F., Collichio, F. and Castillo, M., 2017. Imaging manifestations of pseudoprogression in metastatic melanoma nodes injected with talimogene laherparepvec: initial experience. *American Journal of Neuroradiology*, 38(6), pp.1218-1222.
3. Todo, T., Ino, Y., Ohtsu, H., Shibahara, J. and Tanaka, M., 2022. A phase I/II study of triple-mutated oncolytic herpes virus G47 Δ in patients with progressive glioblastoma. *Nature communications*, 13(1), p.4119.
4. Hansen, A.R., Graham, D.M., Pond, G.R. and Siu, L.L., 2014. Phase 1 trial design: is 3+ 3 the best?. *Cancer control*, 21(3), pp.200-208.
5. Le Tourneau, C., Lee, J.J. and Siu, L.L., 2009. Dose escalation methods in phase I cancer clinical trials. *JNCI: Journal of the National Cancer Institute*, 101(10), pp.708-720.
6. Ivy, S.P., Siu, L.L., Garrett-Mayer, E. and Rubinstein, L., 2010. Approaches to phase 1 clinical trial design focused on safety, efficiency, and selected patient populations: a report from the clinical trial design task force of the national cancer institute investigational drug steering committee. *Clinical Cancer Research*, 16(6), pp.1726-1736.

Response to referees

We are grateful for the opportunity to submit a revised version of our manuscript “Safety, efficacy and biological data from phase 1a of PROTA: A phase I trial combining oncolytic adenovirus TILT-123 with pembrolizumab for the treatment of patients with platinum resistant or refractory ovarian cancer” for further review.

Please find our point by point responses to reviewer comments below.

Reviewer #2 (Remarks to the Author)

- 1. The fact that some patients received intra-peritoneal TILT-123 is a fairly major change that really should have been mentioned clearly in the first version of the manuscript. Why did these patients receive intra-peritoneal rather than intra-tumoural injections? Lines 234-238 detail the treatment regimens but there is still no mention of the doses given by the IP route. Was IP dosing performed via an indwelling catheter?*

AUTHORS RESPONSE

We agree that we understated the inclusion of intraperitoneal (i.p.) injections in the original version of the manuscript, and are grateful for the recommendation to highlight this in the current version.

Three patients enrolled onto the trial presented with ascites, making them eligible for i.p. injections. Two out of the three of these patients receiving i.p. injections also received intra-tumoral (i.t.) injections for other lesions. For example, patient 302-05 received i.p. injections for ascites and two i.t. injections for two metastatic lesions in their lymph nodes. 301-01 who presented with primary peritoneal cancer was the only patient to receive just i.p. injections during the trial, as the patient had no other injectable lesions. We have added the following text to line 262 to further detail the experience of i.p. injections in the trial “301-01 who presented with primary peritoneal cancer was the only patient to receive just i.p. injections during the trial, as the patient had no other injectable lesions. 302-05 and 302-12 received both i.p. and i.t. injections for treating ascites and other metastatic lesions respectively.”

Regarding reporting the i.p. dose, we have now also included the following adjustment on line 241. “for both i.p. and i.t. injections” instead of “for i.t. injections”. We also noted a typo on line 241 in reference to the highest dose provided by i.t. injection, which should be 5×10^{11} rather than 5×10^{12} . This is correctly described in the Methods section, and the typo has been corrected.

The method of i.p. dosing was not indwelling catheter but direct injection into the abdomen using a 22-26 gauge needle. The patient was massaged for 3 minutes in four positions to allow dissemination of the virus in the peritoneal cavity. This has now been added to line 105, Preparation of TILT-123 dosage in Methods section.

2. *Efficacy. I still do not believe that lesion-by-lesion response rates are an appropriate metric and thus I do not believe that the data presented in lines 282-287 are appropriate for a formal clinical study. Response or progression is by patient – thus the efficacy outcome is ORR of 7.1%, rising to 20% at the highest dose level. A simple statement that ‘responses were observed in both injected and non-injected lesions’ is possibly appropriate.*

AUTHORS RESPONSE

As suggested, we have modified the text in lines 282-287 to avoid any confusion between lesion response and RECIST1.1 response rates. As suggested by the reviewer, we have modified the text to the following;

“Treatment related tumor reductions were observed in both injected and non-injected tumors at both CT imaging time points (36 and 92) (Supplementary figure 1b).”

We have also removed the response criteria for lesion-by-lesion analysis from the Assessment of treatment response of the Methods section and the lesion-by-lesion response rates in supplementary figure 1b.

3. *Descriptors such as ‘notably’ (lines 306, 321), notable (line 308), ‘promising’ (line 318) and ‘Remarkably’ (line 321) should be removed: the results are a factual presentation of the data.*

AUTHORS RESPONSE

We agree the results should be presented factually and have now removed these potentially misleading adverbs.

4. *Lines 353-356 state “Interestingly there was a drop in IL-2, hexon and fiber knob expression in the injected lesions from day 36 to 78, whilst expression increased in the non-injected lesions at the same time points. This dynamic likely indicates systemic dissemination of TILT-123 from injected lesions to non-injected tumors.” However, in Figure 3b, there appears to be no measurement of any of these parameters in non-injected lesions after day 36. Thus, I do not believe that this statement is supported by the data presented.*

AUTHORS RESPONSE

We realise the data points for the non-injected tumours were incorrectly input during the revisions. We have now corrected the graphs to reflect the descriptions in the text.

Minor points

1. *Y-axis labels in Figure 4a – “% CD8+, CD45+, PD1+, Epi+” – I think this should be Epi-?? Also what is the denominator in these graphs? All CD45+ cells? All CD8+ cells??*

AUTHORS RESPONSE

The use of Epi+ in our multiplex immunofluorescence data refers more specifically to intraepithelial immune cells, in this case intraepithelial PD-1 expressing CD8 T cells. This is often used as a surrogate for intratumoral immune cells. The denominator of these graphs is in fact the pan epithelial marker and so we have now corrected the y-axis to be %CD8+, CD45+, PD-1+ out of Epi+ cells.

2. *Was CD3 staining also performed to characterise NK cells? CD45+, CD56+ populations will include more than just NK cells (NKT cells, monocytes, dendritic cells etc) and a negative CD3 stain is also required to define NK cells.*

AUTHORS RESPONSE

We understand the limitation of only using CD56 as a marker of NK cells. Indeed CD56 is known to be a marker of NKT cells, monocytes, dendritic cells as well as cancer cells. Since we could not include CD3 in this panel due to the limitation of markers for the available assay set-up, we are not able to further define this population using the proposed marker. For this reason, we have modified the graphs and respective text to refer to "CD56+ positive immune cells" rather than "NK cells".

Reviewer #3 (Remarks to the Author)

1. *Primary (AE) endpoint definition:*

AUTHORS RESPONSE

The following text has now been included in Design of the Methods section on page 4 line 177. (minor: could NOT find such info in the line #s to correspond to the info, but I found them elsewhere)

'AE grades were used to define this AE endpoint, per Adverse Events by NCI-CTCAE version 5.0 and if not possible to capture by this criteria the investigator will use AE criteria described in supplementary figure 4b.'

The primary endpoint remains unclear: what (grades/types) counts as the primary AE endpoint among all the observed AEs, in addition to provide info on the grading method (thanks!).

AUTHORS RESPONSE

We apologise the text/line was incorrectly referred to in the previous round of revisions but we are pleased you could find the new text elsewhere. The primary trial endpoint which is safety by day 92 is based on types, frequency and severity of adverse events. We have now added "types, frequency and severity" onto line 144.

Severity/grading method was included in the previous round of revisions as supplementary figure 4c.

2. *AE summaries:*

The AEs summary (primary endpoint) remain unclear and insufficient. It does not meet the standard of an oncology phase I trial AE reporting and need to be clarified.

Two tables are included to summarize the study AE profile: tab 2 and supplemental 3 (newly added), however,

- Tab 2 is a list of the treatment related AEs types by grades (what grades-maximum?) counts without knowing which cohorts were the events belong to.

- The Supl table 3 is very confusing. It's unclear what the #s mean (unique pts experience AEs?), nor is it consistent with the stated title/legend: there is no maximum grade info (?) and the #s (particularly for cohort 1) doesn't seem to make sense: why would cohort 1 have #s ≥ 3 in the cells (if cohort 1 has only 3 pts, based on table 1)?

- As I noted it before: A tabular summary of counts of AEs by type and grade for each cohort should be included.

AUTHORS RESPONSE

We agree the tables summarising AEs could provide a more comprehensive overview of AEs experienced in the trial. To improve the clarity of the tabular AE summary we have updated table 2 which includes AEs related to treatment stratified by grade, and number of individual (unique) patients experiencing the AE.

Regarding meaning of unique patients experiencing AE. We believe including number of individual (unique) patients that experienced the AE allows the reader to understand interpatient vs inpatient frequency of the AE.

The version of supplementary table 3 provided in the previous round of revisions shows frequency of the maximum grade per cohort. We have now updated the table to show number of patients experiencing AE, percentage in parentheses and maximum grade reported in brackets. We have also added a column reporting total number of TEAEs grade ≥ 3 . We have also updated the AEs according to data cut-off 16/11/24.

3. *Evaluation endpoint: unclear what this is if it seems to refer AE assessment time point at day 36(?)*

AUTHORS RESPONSE

This refers to the two imaging time points performed to evaluate treatment response. We have replaced 'evaluation endpoint' with 'first imaging time point'

In manuscript Line 279: it notes 4/15 patients entered the extension period (cohort?): should it be 4/14 (from evaluation cohort or how was the analysis cohort defined?) Also, in figure 1a, it's still mention of 'evaluation endpoint' (?)

AUTHORS RESPONSE

We agree the figures and text require clarity of wording.

Firstly, in figure 1a we have now corrected "14 patients met evaluation endpoint at day 36" to "14 patients met first imaging time point at day 36" in accordance with the previous revision.

To clarify, the day 36 imaging time point is not specifically an AE assessment time point although information regarding safety is also collected on this day. The imaging time point on day 36 was included to evaluate the tumour response generated by TILT-123 monotherapy.

Additionally, these 4 patient patients did not enter a new cohort together based on an evaluation/cohort analysis. These 4 patients entered the extension treatment period in which patients continued on treatment with the same dose of their respective cohort. Eligibility to continue on treatment is based on if patients show possible beneficial treatment effect in the initial treatment period (up to day 92) and is assessed case-by-case by the sponsor and the clinical investigator. We have incorporated this last sentence onto line 114 in "Design" of "Methods" section in addition to the text described in the response to point 5 below.

4. *Statistical analysis:*

Prior comment: Formal statistical inference such as log-rank test for the comparison of PFS and overall survival OS between groups of disease controls were inappropriate, as the outcomes and the comparison groups are intrinsically correlated. In addition, there was no prior power assessment for such statistical inference.

AUTHORS RESPONSE

As suggested, we have removed the statistical comparisons between the two groups in Figure 1b and c. We have removed the curves for disease control and no disease control to improve clarity and interpretability of Figure 1 for readers.

The section didn't seem to be updated. Is log-rank test still relevant? if so where is being used and why?

AUTHORS RESPONSE

According to our copy of the revised version of the manuscript which was re-submitted, Figure 1b and Figure 1c have been updated and only include curves for median OS and PFS for all patients. The curves for disease control and no disease control are no longer present on the graphs and have been previously omitted from the manuscript.

We have used this statistical method for other analysis within the manuscript, e.g. supplementary figure 1d in which we compare OS and PFS between platinum refractory and resistant patients (if referring to log-rank test which is mentioned in "statistical methods" of "Methods" section). To improve clarity, we have included the following additional information in the description on line 220 in the "statistical methods" of "Methods" section.

"Mantell-Cox Logrank test was used to compare OS and PFS between groups including dose/cohorts, platinum status and serostatus."

5. *What final dose(s) were recommended for the extension period to assess the treatment efficacy?*

AUTHORS RESPONSE

'The phase 1b uses the virus and pembrolizumab dose from cohort 4. The final dose recommended for the next phase of trials evaluating efficacy has yet to be determined as the ongoing phase 1b is not completed yet'

This text has been added to the last lines of the Discussion on page 13 line 581.

However, in fig 1a, there is a box to note that 4 pts are treated in extension cohort, who are they? from cohort 4? Then what (dose combinations?) were they receiving?

Again, the analysis cohort(s) for safety and efficacy endpoints should but was not clearly stated.

AUTHORS RESPONSE

Please note that in Figure 1a the box states "4 treated in extension protocol" which refers to the extension treatment period of the phase 1a not the phase 1b. This is not a new cohort. The extension treatment period is detailed in the treatment schedule of figure 1d and described on line 112 in "Design" of "Methods" section as follows "Patients showing possible beneficial treatment effects in the initial treatment period entered a treatment extension period until completion of up to 38 injections of TILT-123 and 35 cycles of pembrolizumab (up to approximately 2 years)."

To improve clarity we have modified "4 treated in extension protocol" to "4 treated in extension treatment period". Additionally we have listed the patient ID of the four patients (301-11, 301-05, 302-06 and 302-10) which entered the extension treatment period in the same box of Figure 1a and included which cohort they belonged to in parentheses. We have also updated the respective text on line 279 in "Efficacy" of "Results" section. "The overall response rate (ORR) was 7.1% (1/14) and 20% (1/5) at the highest dose level. 4/15 (301-11, 301-05, 302-06 and 302-10) patients entered the extension period and received a total of 22 additional treatments between them."

We have now also detailed that patients will remain in the same dose cohort when entering the extension treatment period on line 112 in "Design" of "Methods" section as follows.

"Patients showing possible beneficial treatment effects in the initial treatment period entered an extension treatment period (continuing on the same respective dose as initial treatment period) until completion of up to 38 injections of TILT-123 and 35 cycles of pembrolizumab (up to approximately 2 years)."

The phase 1b is still enrolling patients and the statement referring to determination of dose for efficacy evaluation is still valid. We also mentioned on line 297 in "Efficacy" of "Results" section "No significant difference between individual cohorts was observed when using Mantell-Cox Logrank test (Supplementary figure 1c)."

6. Results presentation:

- a) Patient characteristics: There is no mention of who were in the extension cohort (which is noted n=4 from fig 1a). Are they a part of those pts enrolled in the (4) cohorts or a separate group?

AUTHORS RESPONSE

As mentioned in the previous comment we have now implemented changes to figure 1a and corresponding texts to improve clarity of which patient's entered the extension treatment period.

- b) Tab 1 and supl tab 1 have some overlap/redundant info (e.g, Histological subtype, Best overall response per RECIST1.1 or per IRECIST respectively), are they necessary? One summary tab that could be helpful (but missing) is the aggregated pt (demographic/disease) characteristics according to the dose cohorts.

AUTHORS RESPONSE

The overlapping information was included to allow the reader to more easily gauge potential association between disease status and treatment outcome. However we agree that this repeated information can be considered redundant. We have therefore removed the repeated information (Histological type, best overall response per RECIST/iRECIST and included dose cohort (although stated in table 1), ascites status and Laterality of the disease (only other available disease related feature not currently included in the manuscript).

- c) As I noted above, the AEs (primary endpoint) summary is unclear and inadequate and should be clarified.

AUTHORS RESPONSE

As mentioned in the previous comment (point 2 above) we have now updated the AE tabular summary to improve clarity.

- d) Efficacy results (line 289-292): Is this relevant or should it be updated? Who are included in this analysis (cohort)? What/why tests/p-values, what are the proper interpretations? e.g, Line 289: what does it mean: 'Median OS was longer in the higher dose cohorts but not significantly'?

AUTHORS RESPONSE

The results described on line 289-295 refer to analysis of overall survival between cohorts 1+2 vs 3+4 using Mantell-Cox Logrank test. Previous analysis showed no statistically significant difference in overall survival when comparing individual cohorts. We therefore decided to proceed with low (1+2) vs high (3+4) cohort analysis. The result of this analysis demonstrated a median overall survival of 107.5 and 280 days for low and high cohorts respectively with a p-value of 0.1726 when using using Mantell-Cox Logrank test. We did also calculate using Grehan-Breslow-Wilcoxon test which provided a p value of 0.0899. Regardless the result was not significant but we deemed the non-significant trend still worth reporting.

However, we agree the current description on line 289-295 in "Efficacy" of "Results" and respective figure requires additional clarification and improvement. We have therefore included the results of the statistical analysis in supplementary 1c but also included an additional graph in which cohorts 1+2 vs 3+4 are shown as supplementary figure 1d.

We have also modified the text to as follows.

“Analysis of OS according to dose/cohort revealed a median overall survival of 85, 122, 280 and 202.5 days for cohort 1, 2, 3 and 4 respectively. No significant difference between individual cohorts was observed when using Mantell-Cox Logrank test (supplementary figure 1c). When groups were compared according to cohorts 1 and 2 vs 3 and 4 (lower vs higher), median overall survival was 107.5 days and 280 days respectively, however no statistical significance (P=0.1726) was achieved (supplementary figure 1d). The non-significant trend observed in individual cohort analysis and grouped cohorts suggests longer overall survival when using the dose of either cohort 3 or 4”

- e) A few typos/errors: I likely missed some. Please cross check the content with tabs/figs.
- a. Figure 1a: Pg 6 line 238 states 6/14 met the primary endpoint at day 92, while figure 1a states 7/14 met the primary endpoint. What's the analysis cohort for the efficacy endpoint?

AUTHORS RESPONSE

We have now corrected the typos on line 238 (now line 241) and updated the figure 1a to 7/14 patients meeting the primary endpoint according to the updated eCRF. The analysis cohort for the efficacy endpoint includes the 14 patients which having imaging data available as shown in Table 1. 302-11 was not imaged and was not included in the analysis. The latter has been added to Figure 1a.

- b. Line 268/pg7: Supplemental figure table 4 should be 'supplemental table 4'

AUTHORS RESPONSE

We have now corrected the typos on line 268/pg7 and cross checked the contents with tabs/figures as suggested. This includes updating the all data according to the updated eCRF.

Response to referees

We are grateful for the opportunity to submit a revised version of our manuscript now named “The oncolytic adenovirus TILT-123 with pembrolizumab in platinum resistant or refractory ovarian cancer: the phase 1a PROTA trial” for further review.

Please find our point by point responses to reviewer comments below.

Reviewer #2 (Remarks to the Author)

- One small suggestion: on lines 92-93, the inclusion criteria state 'At least one tumor (>14 mm in diameter) or carcinomatosis must be available for local virus injection (i.t. and/or i.p.)' This is ambiguous. I would suggest that they state that 'at least one tumor >14mm was required for i.t. injection. In patients with carcinomatosis not amenable to i.t. injection, i.p. injection was used', or something similar.*

AUTHORS RESPONSE

Thank you for your final suggestion, we have incorporated your wording into our text as stated “At least one tumor >14mm was required for i.t. injection. In patients with carcinomatosis not amendable to i.t. injection, i.p. injection was used’.

Reviewer #2 (Remarks to the Author)

- Thanks for the thorough revisions and addressing my prior comments/questions. I have no further questions.*

AUTHORS RESPONSE

We thank reviewer 3 for providing thorough feedback.